# Exponential-Wrapped Mechanisms for Differential Privacy on Hadamard Manifolds

## Abstract

We extend the Differential Privacy (DP) framework to Hadamard manifolds, the class of complete and simply connected Riemannian manifolds with non-positive sectional curvature. Inspired by the Cartan–Hadamard theorem, we introduce Exponential-Wrapped Laplace and Gaussian mechanisms to achieve $\varepsilon$-DP, $(\varepsilon, \delta)$-DP, Gaussian DP (GDP), and Rényi DP (RDP) on these manifolds. Our approach employs efficient, straightforward algorithms that circumvent the computationally intensity Monte Carlo Markov Chain (MCMC) methods. This work is the first to extend $(\varepsilon, \delta)$-DP, GDP, and RDP to Hadamard manifolds. We further demonstrate the effectiveness of our methodology through simulations on the space of Symmetric Positive Definite Matrices, a frequently used Hadamard manifold in statistics. Our findings reveal that our Exponential-Wrapped mechanisms surpass traditional MCMC-based approaches, which require careful tuning and extensive diagnostics, in both performance and ease of use. Additionally, our methods achieve comparable utility to the Riemannian Laplace mechanism with enhanced utility for smaller privacy budgets ($\varepsilon$) and operate orders of magnitude faster computationally.

## 1 Introduction

The proliferation of AI and machine learning technologies has catalyzed the exploration of more complex types of data. Notably, nonlinear manifold data, which frequently emerge in fields such as medical imaging (Pennec et al., 2019; Dryden, 2005; Dryden et al., 2009), computer vision (Turaga & Srivastava, 2015; Turaga et al., 2008; Cheng & Vemuri, 2013), pattern recognition (Nielsen, 2013; Hettiarachchi & Peters, 2015), signal processing (Barachant et al., 2010; Zanini et al., 2018), and geometric deep learning (Belkin et al., 2006; Niyogi, 2013), pose unique challenges.

As data becomes increasingly complex, the task of safeguarding privacy also becomes more challenging and intricate. Differential Privacy (DP) (Dwork et al., 2006b), a leading mathematical framework, has been widely recognized for its ability to quantify and ensure privacy protection. While numerous mechanisms have been developed to achieve DP (McSherry & Talwar, 2007; Barak et al., 2007; Wasserman & Zhou, 2010; Reimherr & Awan, 2019), these traditional mechanisms, primarily designed for linear data, often fall short when applied to complex nonlinear data. For instance, the commonly adopted extrinsic method embeds nonlinear data into Euclidean space, allowing the application of standard differential privacy (DP) mechanisms. However, as Reimherr et al. (2021) demonstrated, leveraging the intrinsic properties of nonlinear data can significantly enhance data utility while maintaining privacy. This underscores the necessity for privacy mechanisms that integrate differential geometry to effectively address the complexities of nonlinear data and fully leverage its geometric structure.

The differential privacy framework was first extended to general manifolds by Reimherr et al. (2021), who introduced the Riemannian Laplace mechanism on Riemannian manifolds to achieve $\varepsilon$-DP. However, the development of other differential privacy variations for broader manifold applications remains limited. Utpala et al. (2023a) extended $(\varepsilon, \delta)$-DP only to a specific manifold – the Symmetric Positive Definite Matrices (SPDM) space – using the log-Euclidean metric instead of the usual Rao-Fisher affine invariant metric. When equipped with the log-Euclidean metric, the SPDM space becomes geometrically flat (Arsigny et al., 2007), which simplifies the approach but at the expense of generality. In a similar vein, Jiang et al. (2023) expanded Gaussian Differential Privacy (GDP) to general manifolds, although their calibration algorithm is confined to constant curvature spaces

and demands considerable computational resources. The sampling methods required for both the Riemannian Laplace and Gaussian mechanisms, as described by Reimherr et al. (2021) and Jiang et al. (2023), involve computationally intensive Markov Chain Monte Carlo (MCMC) processes, particularly in complex, high-dimensional spaces such as the SPDM space. This highlights an ongoing need for broader extensions of differential privacy variations to manifolds and the creation of more computationally efficient mechanisms.

In this research, we significantly advance the application of differential privacy to Hadamard Riemannian manifolds, showcasing robust mechanisms alongside empirical validations. Our main contributions are outlined below::

- We are the first to extend $(\varepsilon, \delta)$-DP, GDP, and RDP to Hadamard Riemannian manifolds through the introduction of Exponential-Wrapped Laplace and Gaussian mechanisms. Importantly, this is the first extension for RDP and broadens the scope of privacy frameworks to general manifold settings.
- We develop fast and efficient implementations of these mechanisms that avoid the computationally intensive MCMC sampling methods. This development facilitates more feasible and practical applications of differential privacy in real-world scenarios.
- Through comprehensive numerical experiments, our results demonstrate that our mechanisms perform comparably to the traditional Riemannian Laplace mechanism. Notably, when achieving GDP, our Exponential Gaussian mechanism exhibits superior performance in scenarios with small privacy budgets.

This paper is structured as follows: First, we review key concepts from Riemannian Geometry and Differential Privacy. Subsequently, we introduce the Exponential-Wrapped Distribution and detail its calibration to achieve $(\varepsilon, \delta)$-DP, GDP, and RDP. We then explore the task of releasing differentially private Fréchet means and derive theoretical utility bounds for our mechanisms. Finally, we present numerical simulations to demonstrate the effectiveness of our methods.

## 2 BACKGROUND MATERIALS

We begin by outlining core concepts in Riemannian Geometry, with reference to standard texts such as Lee (2006); Petersen (2006); Pennec et al. (2019); Said (2021); Grigoryan (2009). Following this, we examine important definitions and results related to DP, GDP, and Rényi DP. For those seeking a deeper understanding, see Dwork & Roth (2014); Mironov (2017); Dong et al. (2021; 2022) for comprehensive discussions.

### 2.1 RIEMANNIAN GEOMETRY

Let $\mathcal{M}$ denote a $d$-dimensional Riemannian manifold endowed with a Riemannian metric $g$, which consists of a smoothly varying collection of inner products $\langle \cdot, \cdot \rangle_x$ defined on each tangent space $T_x\mathcal{M}$ at points $x$ on the manifold. At each point $x$, the inner product is a positive definite bilinear map $\langle \cdot, \cdot \rangle_x : T_x\mathcal{M} \times T_x\mathcal{M} \to \mathbb{R}$. It follows that a norm $\|\cdot\|_x : T_x\mathcal{M} \to \mathbb{R}$ is induced by $\|v\|_x = \langle v, v \rangle_x^{1/2}$. The Riemannian metric $g$ lets us define length and distance on $\mathcal{M}$. Consider a smooth curve $\gamma(t)$ on $\mathcal{M}$, its length is defined by the integral

$$L(\gamma) = \int \|\dot{\gamma}(t)\|_{\gamma(t)} dt = \int \sqrt{\langle \dot{\gamma}(t), \dot{\gamma}(t) \rangle_{\gamma(t)}} \, dt$$

where the $\dot{\gamma}(t)$ is the velocity vector and the integral is over the domain of the curve $\gamma(t)$. Proceeding from this, the distance between two points $x, y \in \mathcal{M}$ is defined as the infimum of the lengths of all piece-wise smooth curves from $x$ to $y$, $d(x, y) = \inf_{\gamma(0)=x, \gamma(1)=y} L(\gamma)$. Lastly, we introduce the concept of a measure on $\mathcal{M}$. In any chart $U$, the Riemannian metric $g$ can be represented by the matrix $g = (g_{ij})$, and the Lebesgue measure is denoted by $\lambda$. The metric $g$ induces a unique measure $\nu$ on the Borel $\sigma$-algebra of $\mathcal{M}$, such that $d\nu = \sqrt{\det g} d\lambda$.

In Riemannian manifolds, curves that locally minimize length are referred to as geodesics. A Riemannian manifold $\mathcal{M}$ is called geodesically complete if the domain of all geodesics can be extended to $\mathbb{R}$. From now on, $\mathcal{M}$ is assumed to be geodesically complete. Consider a point $p \in \mathcal{M}$

and a tangent vector $v \in T_p\mathcal{M}$. There exists a unique geodesic $\gamma_{(p,v)}(t)$ starting from $p = \gamma_{(p,v)}(0)$ with tangent vector $v = \dot{\gamma}_{(p,v)}(0)$, which is defined initially only in a small neighborhood of zero, but can be extended to $\mathbb{R}$ due to the geodesics completeness assumption. This leads to the definition of the exponential map $\mathrm{Exp}_p : T_p\mathcal{M} \to \mathcal{M}$ as $\mathrm{Exp}_p(v) = \gamma_{(p,v)}(1)$. Furthermore, there exists a neighborhood $V$ of the origin in $T_p\mathcal{M}$ and a neighborhood $U$ of $p$ such that the restriction $\mathrm{Exp}_p|_V : V \to U$ is a diffeomorphism. Within the neighborhood $U$, we can define the inverse of $\mathrm{Exp}_p$ as $\mathrm{Log}_p(q) = v$ where $q = \gamma_{(p,v)}(1)$. Additionally, we have $d(p,q) = \|\mathrm{Log}_p(q)\|_p$.

The primary focus of this paper is on Hadamard manifolds. *A simply connected complete Riemannian manifold of non-positive curvature* is called a **Hadamard Manifold**. It is named after the famous Cartan-Hadamard theorem which states that for any $d$-dimensional Hadamard manifold $\mathcal{M}$, it is differomorphic to $\mathbb{R}^d$ and more precisely, at any point $p \in \mathcal{M}$, the exponential mapping $\mathrm{Exp}_p : T_p\mathcal{M} \to \mathcal{M}$ is a diffeomorphism and thus $\mathrm{Log}_p$ is defined everywhere on $\mathcal{M}$. This property enables us to develop the Exponential-Wrapped mechanisms in Sections 3.2 and 3.3. Another important property of the Hadamard manifold is that $\mathrm{Log}_p$ is a contraction for any $p \in \mathcal{M}$. That is, $\|\mathrm{Log}_p q_1 - \mathrm{Log}_p q_2\| \leq d(q_1, q_2)$ for any $p, q_1, q_2 \in \mathcal{M}$. For more technical details on Hadamard manifolds, please refer to Petersen (2006); Shiga (1984).

## 2.2 DIFFERENTIAL PRIVACY

**Definition 2.1** ((Dwork et al., 2006a)). *A data-releasing mechanism $M$ is said to be $(\varepsilon, \delta)$-DP with $\varepsilon \geq 0, 0 \leq \delta \leq 1$, if for any adjacent datasets, denoted as $\mathcal{D} \simeq \mathcal{D}'$, differing in only one record, we have $\Pr(M(\mathcal{D}) \in A) \leq e^\varepsilon \Pr(M(\mathcal{D}') \in A) + \delta$ for any measurable set $A$ in the range of $M$. For $\delta = 0$, $M$ is said to be $\varepsilon$-DP*.

Since $(\varepsilon, \delta)$-DP is a well-defined concept on any measurable space (Wasserman & Zhou, 2010), it can be readily extended to any Riemannian manifold equipped with the Borel $\sigma$-algebra.

One relaxation of $\varepsilon$-DP is the Rényi DP, which is based on Rényi divergence. It shares many important properties with $\varepsilon$-DP while allowing tighter analysis of composite heterogeneous mechanisms.

**Definition 2.2** ((Mironov, 2017)). *A mechanism $M$ is said to have $\epsilon$-**Rényi Differential Privacy (RDP)** of order $\alpha$, or $(\alpha, \epsilon)$-RDP for short, if $D_\alpha(M(\mathcal{D})\|M(\mathcal{D}')) \leq \epsilon$ for all neighboring datasets $D \simeq D'$, where the Rényi divergence of a finite order $\alpha \neq 1$ is defined as*

$$D_\alpha(P\|Q) = \frac{1}{\alpha - 1} \log \mathbb{E}_{x \sim Q} \left(\frac{P(x)}{Q(x)}\right)^\alpha,$$

*and Renyi divergence at orders $\alpha = 1, \infty$ are defined by continuity.*

Another way of extending the differential privacy definition is through the viewpoint of the statistical hypothesis testing (Wasserman & Zhou, 2010; Kairouz et al., 2017). In the context of hypothesis testing, we define $H_0$: the underlying dataset is $\mathcal{D}$ and $H_1$: the underlying dataset is $\mathcal{D}'$. As the values of $\varepsilon$ and $\delta$ decrease, the task of conducting this hypothesis testing becomes more difficult. This means that detecting the presence of an individual based on the outcome of the mechanism becomes increasingly challenging. With this interpretation in mind, we can extend $(\varepsilon, \delta)$-differential privacy to Gaussian differential privacy (GDP). Denote the outcome distribution under $H_0$ and $H_1$ as $M(\mathcal{D})$ and $M(\mathcal{D}')$, respectively. We introduce the optimal tradeoff function beween type I and type II errors as follow,

$$T(M(\mathcal{D}), M(\mathcal{D}')) : [0,1] \to [0,1], \alpha \mapsto T(M(\mathcal{D}), M(\mathcal{D}'))(\alpha)$$

where $T(M(\mathcal{D}), M(\mathcal{D}'))(\alpha)$ is the smallest type II error when type I error equals $\alpha$. GDP centers around this optimal tradeoff function and is defined as follow.

**Definition 2.3** ((Dong et al., 2022)). *A mechanism $M$ is said to satisfy $\mu$-**Gaussian Differential Privacy** ($\mu$-**GDP**) if $T(M(\mathcal{D}), M(\mathcal{D}')) \geq G_\mu$ for all neighboring datasets $\mathcal{D} \simeq \mathcal{D}'$ with $G_\mu := T(N(0,1), N(\mu,1))$.*

However, the involvement of the optimal trade-off function $T(M(\mathcal{D}), M(\mathcal{D}'))$ makes Definition 2.3 difficult to work with on Riemannian manifolds. Instead, we adapt the GDP definition from Jiang et al. (2023), which is based on Corollary 1 from (Dong et al., 2022).

**Definition 2.4** (Gaussian Differential Privacy (Dong et al., 2022; Jiang et al., 2023))**.** *A $\mathcal{M}$-valued data-releasing mechanism $M$ is said to be $\mu$-**GDP** if it's $(\varepsilon, \delta_\mu(\varepsilon))$-DP for all $\varepsilon \geq 0$, where*

$$\delta_\mu(\varepsilon) := \Phi\left(-\frac{\varepsilon}{\mu} + \frac{\mu}{2}\right) - e^\varepsilon \Phi\left(-\frac{\varepsilon}{\mu} - \frac{\mu}{2}\right).$$

*and $\Phi$ denotes the cumulative distribution function of the standard normal distribution.*

## 3 Differential Privacy on Hadamard Riemannian Manifolds

In this section, we outline the key idea of this study, which revolves around the idea of Exponential-Wrapped Distribution. Section 3.1 provides the necessary background on Exponential-Wrapped Distributions. Following this, Section 3.2, introduces the Exponential-Wrapped Laplace Mechanism which is calibrated to achieve $\varepsilon$-DP. In Section 3.3, we present the Exponential-Wrapped Gaussian Mechanism, calibrated to achieve $(\varepsilon, \delta)$-DP, GDP, and RDP.

### 3.1 Exponential-Wrapped Distribution

In measure-theoretic terms, the Exponential-Wrapped Probability is the push-forward of the tangent space probability via the exponential map. For manifold $\mathcal{M}$ with dimension $d > 1$, wrapping a density around the manifold involves volume distortion. This occurs because the exponential map typically does not preserve the area between the Lebesgue measure on the tangent space and the reference measure on the manifold.

Let $\mathcal{M}$ be a manifold with a base measure $\nu$. Given $\mu$, a probability distribution on $T_p\mathcal{M}$ with a probability density $h$ w.r.t the Lebegue measure $\lambda_p$ on $T_p\mathcal{M}$, the corresponding exponential-wrapped distribution is defined as the push-forward of $\mu$ by the exponential, $\Lambda = \mathrm{Exp}_{p*}\mu$, where the $*$ refers to the push-forward by $\mathrm{Exp}_p$ such that $\Lambda(A) = \mu\left(\mathrm{Log}_p(A)\right)$. Since we assume $\mathcal{M}$ is a Hadamard manifold, $\mathrm{Log}_p$ is defined everywhere on $\mathcal{M}$ for any $p \in \mathcal{M}$. If follows that the density $g$ of $\Lambda$ can be expressed from $h$ and a volume change term,

$$g(q) = \frac{\mathrm{d}\Lambda}{\mathrm{d}\nu}(q) = \frac{\mathrm{d}\,\mathrm{Exp}_{p*}(\lambda_p)}{\mathrm{d}\nu}\frac{\mathrm{d}\Lambda}{\mathrm{d}\,\mathrm{Exp}_{p*}(\lambda_p)}(q) = \frac{\mathrm{d}\,\mathrm{Exp}_{p*}(\lambda_p)}{\mathrm{d}\nu}(q)h(\mathrm{Log}_p q) = \frac{h(\mathrm{Log}_p q)}{J_p(\mathrm{Log}_p(q))},$$

where $J_p$ is the Jacobian determinant of the exponential map.

The most attractive property of the Exponential-Wrapped Distribution is its straightforward sampling procedure. In order to sample from $g$, it suffices to sample from $h$: if $U_1, \ldots, U_n$ are i.i.d. random variables on a tangent space $T_p\mathcal{M}$ following the density $h$, then $X_1 = \mathrm{Exp}_p(U_1), \ldots, X_n = \mathrm{Exp}_p(U_n)$ are i.i.d. random variables on $\mathcal{M}$ following the density $g$. For a more detailed discussion on Exponential-Wrapped Distribution, please refer to Chevallier et al. (2022).

### 3.2 Exponential-Wrapped Laplace Mechanism

Cetin et al. (2023)

**Definition 3.1** (Exponential-Wrapped Laplace Distribution)**.** *Let $\mathcal{M}$ be a Hadamard Riemannian manifold with measure $\nu$, we define a probability density function w.r.t $\nu$ as*

$$g(y) \propto \frac{1}{J_{p_0}(\mathrm{Log}_{p_0}(y))} \exp\left(-\frac{\|\mathrm{Log}_{p_0}(y) - \mathrm{Log}_{p_0}\eta\|}{\sigma}\right),$$

*We called this distribution an **Exponential-Wrapped Laplace Distribution** with **footpoint** $p_0$, **center** $\eta$ and **rate** $\sigma > 0$.*

The Exponential-Wrapped Laplace Distribution is the push-forward probability of the tangent space probability defined by the probability density $h(u) \propto \exp\{-\|u - \mathrm{Log}_{p_0}\eta\|/\sigma\}$. We present the following theorem to demonstrate how it can be used to achieve $\varepsilon$-DP.

**Theorem 3.1** (Exponential-Wrapped Laplace Mechanism)**.** *Let $\mathcal{M}$ be a Hadamard Riemannian manifold and $f$ be a $\mathcal{M}$-valued summary with global sensitivity[1] $\Delta$. The Exponential-Wrapped Laplace Distribution with footprint $p_0$, center $f(\mathcal{D})$ and rate $\Delta/\varepsilon$ satisfies $\varepsilon$-DP.*

---

[1] A summary $f$ is said to have a **global sensitivity** of $\Delta < \infty$, with respect to $d(\cdot, \cdot)$, if we have $d\left(f(\mathcal{D}), f(\mathcal{D}')\right) \leq \Delta$ for any two datasets $\mathcal{D} \simeq \mathcal{D}'$.

*Proof.* See Appendix A.1.1. □

Compared to the Riemannian Laplace mechanism proposed by Reimherr et al. (2021), the Exponential-Wrapped Laplace mechanism defined above offers two primary advantages. First, our method only requires a rate of $\Delta/\varepsilon$ to achieve $\varepsilon$-DP across all manifolds, homogeneous or not. This is more efficient than the Riemannian Laplace mechanism, which necessitates a rate of $2\Delta/\varepsilon$ for non-homogeneous manifolds. Second, our approach is easier to implement and less computationally complex. The Riemannian Laplace mechanism relies on MCMC sampling, which is computationally intensive due to prolonged burn-in iterations and frequent recalculations of Riemannian distances. These computations escalate in cost with increasing manifold dimensionality. Even in SPDM space with the Rao-Fisher affine invariant metric, where efficient sampling techniques for the Riemannian Laplace Distribution exist (Hajri et al., 2016) – MCMC procedures remain necessary, and the choice of proposal distribution critically affects convergence. In contrast, sampling from the Exponential-Wrapped Laplace Distribution is straightforward: it involves 1) sampling from $u \sim h(u) \propto \exp\{-\|u - \text{Log}_{p_0} \eta\|/\sigma\}$ and 2) computing $\text{Exp}_{p_0} u$. The complete algorithm is detailed in Algorithm 1.

---

**Algorithm 1** Exponential-Wrapped Laplace Mechanism for $\varepsilon$-DP

**Input:** Sensitivity $\Delta$, privacy budget $\varepsilon$, query result $f(\mathcal{D})$, footpoint $p_0$.
**Output:** Privatized query result $\tilde{f}(\mathcal{D})$
  1: **Sample** $v$ uniformly from $\mathbb{S}^{d-1}$ and $r$ from Gamma distribution $\Gamma(d, 1)$.[2]
  2: **Compute** $u = \text{Log}_{p_0} f(\mathcal{D}) + rv\sigma$ and $\tilde{f}(\mathcal{D}) = \text{Exp}_{p_0} u$.
  3: **Return**: $\tilde{f}(\mathcal{D})$.

---

**Remark 1.** *Note that there is no restriction on the choice of footpoint $p_0$ in the exponential-wrapped Laplace mechanism. However, its selection can have an impact on the performance of the mechanism. Furthermore, to be compliant with the differential privacy definition, the selection of the footpoint $p_0$ cannot be based on the private dataset $\mathcal{D}$. For more discussion on the selection of footpoint, see Section 4.2.*

### 3.3 Exponential-Wrapped Gaussian Mechanism

Beyond the Laplace mechanism, the Gaussian mechanism stands as one of the most prevalent tools in DP (Dwork & Roth, 2014; Balle & Wang, 2018). This section introduces the Exponential-Wrapped Gaussian mechanism, calibrated to achieve $(\varepsilon, \delta)$-DP, RDP, and GDP. Initially, we will define the Exponential-Wrapped Gaussian Distribution as follows.

**Definition 3.2** (Exponential-Wrapped Gaussian Distribution)**.** *Let $\mathcal{M}$ be a Hadamard Riemannian manifold with reference measure denoted by vol, we define a probability density function w.r.t vol as*

$$g(y) \propto \frac{1}{J_{p_0}(\text{Log}_{p_0}(y))} \exp\left(-\frac{\|\text{Log}_{p_0}(y) - \text{Log}_{p_0} \eta\|^2}{2\sigma^2}\right).$$

*We called this distribution an **Exponential-Wrapped Gaussian Distribution** with footpoint $p_0$, tangent center $\eta$, and rate $\sigma > 0$.*

The Exponential-Wrapped Gaussian Distribution is defined as the push-forward of the multivariate Gaussian distribution, characterized by a mean of $\text{Log}_{p_0} \eta$ and a covariance of $\sigma^2 \mathbf{I}$, on the tangent space $T_p\mathcal{M}$. We present the following theorem to demonstrate how it can be used to achieve $(\varepsilon, \delta)$-DP.

**Theorem 3.2.** *Let $\mathcal{M}$ be a Hadamard Riemannian manifold and $f$ be a $\mathcal{M}$-valued summary. The Exponential-Wrapped Gaussian Distribution with footprint $p_0$, tangent center $\text{Log}_{p_0} f(\mathcal{D})$ and rate $\sigma$ satisfies $(\varepsilon, \delta)$-DP if and only if the following condition is satisfied,*

$$\Phi\left(-\frac{\sigma\varepsilon}{\Delta_{p_0}} + \frac{\Delta_{p_0}}{2\sigma}\right) - e^{\varepsilon}\Phi\left(-\frac{\sigma\varepsilon}{\Delta_{p_0}} + \frac{\Delta_{p_0}}{2\sigma}\right) \leq \delta \tag{1}$$

*where $\Delta_{p_0} = \sup_{D \simeq D'} \|\text{Log}_{p_0}(f(\mathcal{D})) - \text{Log}_{p_0}(f(\mathcal{D}'))\|.$*

---

[2]Note that the purpose of step 1 is to sample a random variable with a density propositional to $\exp(-\|\cdot\|)$. For more details, refer to Appendix 1.4 of Reimherr et al. (2021).

*Proof.* See Appendix A.1.2. □

Theorem 3.2 shares similarities with the analytic Gaussian mechanism in Balle & Wang (2018). Primary distinction lies in the use of $\Delta_{p_0}$ rather than the standard sensitivity $\Delta$ in inequality (1). This substitution generally does not pose significant challenges; if $\Delta_{p_0}$ proves difficult to compute, $\Delta$ can be use instead in (1) as $\Delta \geq \Delta_{p_0}$ since $\text{Log}_{p_0}$ is a contraction for Hadamard manifolds.

Implementing the Exponential-Wrapped Gaussian mechanism for $(\varepsilon, \delta)$-DP is straightforward. After determining the appropriate $\sigma$ numerically from inequality (1)—using a method such as that proposed in Balle & Wang (2018)—one can proceed by first sampling $\mathbf{u}$ from the multivariate Gaussian distribution $\mathbf{N}d(\mathbf{0}, \sigma^2\mathbf{I}d)$. The privatized summary is then computed as $\text{Exp}\, p_0(\mathbf{u} + \text{Log}\, p_0(f(\mathcal{D})))$.

Suppose $\mathcal{M}$ is the space of SPDM equipped with log-Euclidean metric, the Exponential-Wrapped Gaussian mechanism with footprint $p_0 = \mathbf{I}$ reduces to the tangent Gaussian mechanism in Utpala et al. (2023b). Hence, the Exponential-Wrapped Gaussian mechanism is a generalization of the tangent Gaussian mechanism as our mechanism can be employed for any Hadamard manifolds equipped with any Riemannian metric.

Similar to how the Euclidean Gaussian Distribution can be used to achieve GDP, we can calibrate the Exponential-Wrapped Gaussian Distribution to achieve GDP in the following manner.

**Theorem 3.3** (Wrapped Gaussian Mechanism for GDP). *Let $\mathcal{M}$ be a Hadamard Riemannian manifold and $f$ be a $\mathcal{M}$-valued summary with global sensitivity $\Delta$. The Exponential-Wrapped Gaussian Distribution with footprint $p_0$, tangent center $\text{Log}_{p_0} f(\mathcal{D})$ and rate $\Delta/\mu$ satisfies $\mu$-GDP.*

*Proof.* See Appendix A.1.3. □

Previously, Jiang et al. (2023) introduced the Riemannian Gaussian mechanism to achieve $\mu$-GDP. However, our approach presents significant advantages in both calibration and sampling. Firstly, the Riemannian Gaussian mechanism requires the resolution of infinitely many integral inequalities to calibrate the rate $\sigma$ for a given privacy budget $\mu$. The calibration algorithm provided by Jiang et al. (2023) is only applicable to constant curvature spaces and is computationally intensive, involving grid searches and MCMC techniques to compute the integrals. In contrast, our method simplifies calibration to a straightforward calculation: $\sigma = \Delta/\mu$. Secondly, like the Riemannian Laplace distribution, sampling from the Riemannian Gaussian distribution involves complex processes (detailed in section 3.2). Our sampling technique is considerably simpler, requiring only the sampling from a multivariate Gaussian distribution followed by computations using $\text{Exp}\, p_0$ and $\text{Log}\, p_0$.

In a similar fashion, we can use Exponential-Wrapped Gaussian Distribution to achieve RDP.

**Theorem 3.4** (Wrapped Gaussian Mechanism for Rényi DP). *Let $\mathcal{M}$ be a Hadamard Riemannian manifold and $f$ be a $\mathcal{M}$-valued summary with global sensitivity $\Delta$. The Exponential-wrapped Gaussian distribution with footprint $p_0$, tangent center $\text{Log}_{p_0} f(\mathcal{D})$ and rate $\Delta/\sqrt{2\varepsilon/\alpha}$ satisfies $(\alpha, \varepsilon)$-RDP.*

*Proof.* See Appendix A.1.4. □

## 4 DIFFERENTIALLY PRIVATE FRÉCHET MEAN AND UTILITY GUARANTEE

In Section 4.1, we discuss the task of releasing differentially private Fréchet mean, a topic extensively discussed in the literature on differential privacy over manifolds (Reimherr et al., 2021; Soto et al., 2022; Utpala et al., 2023b). Section 4.2 details the derivation of theoretical utility bounds for our Exponential-Wrapped mechanisms for releasing a private Fréchet mean.

### 4.1 DIFFERENTIALLY PRIVATE FRÉCHET MEAN

For a comprehensive overview of the Fréchet mean in the context of DP, please refer to Reimherr et al. (2021). Consider a set of data $x_1, \ldots, x_N$ on the manifold $\mathcal{M}$. The Euclidean sample mean can be generalized to Riemannian manifolds as the sample Fréchet mean, w defined as the minimizer of the sum-of-squared distances to the data points, $\bar{x} = \arg\min_{x \in \mathcal{M}} \sum_{i=1}^{N} d(x, x_i)^2$. The properties

of Hadamard manifolds guarantee the existence and uniqueness of the Fréchet mean. To ensure the sensitivity of the sample Fréchet mean is finite and properly decreasing with sample size, we need the following assumption:

**Assumption 1.** *The data $\mathcal{D} \subseteq B_r(m_0)$ for some $m_0 \in \mathcal{M}$, $r < \infty$.*

The assumption that data lies within a bounded ball is standard in the field of DP and should not raise concerns. Consider two datasets $\mathcal{D} \simeq \mathcal{D}'$ on $\mathcal{M}$, and denote $\bar{x}$ and $\bar{x}'$ as the two sample Fréchet means of $\mathcal{D}$ and $\mathcal{D}'$ respectively. Under Assumption 1, we have $d(\bar{x}, \bar{x}') \leq 2r/n$.

### 4.2 UTILITY GUARANTEE

To evaluate the theoretical utility, we derive bounds on the expected distance between the output of our mechanism and the true sample Fréchet mean $F(\mathcal{D})$.

**Theorem 4.1.** *Let $\mathcal{M}$ be a $d$-dimensional Hadamard manifold and assume assumption 1 holds. Denote $\tilde{x}_{EWL}$ as a sample drawn from an Exponential-Wrapped Laplace Distribution with footprint $p_0$, tangent center $\mathrm{Log}_{p_0} f(\mathcal{D})$ and rate $\sigma = \Delta/\varepsilon$. $\tilde{x}_{EWL}$ is $\varepsilon$-DP and we have*

$$\mathbb{E}\, d(\tilde{x}_{EWL}, \bar{x}) \leq \sigma d + 2d(p_0, f(\mathcal{D})). \tag{2}$$

*With $p_0 = m_0$ and $\sigma \leq 2r/(n\varepsilon)$, we have,*

$$\mathbb{E}\, d(\tilde{x}_{EWL}, \bar{x}) \leq \frac{2r}{n\varepsilon} d + 2r.$$

*Similarly, denote $\tilde{x}_{EWG}$ as a sample drawn from an Exponential-Wrapped Gaussian Distribution with footprint $p_0$, tangent center $\mathrm{Log}_{p_0} f(\mathcal{D})$ and rate $\sigma = \Delta/\mu$. $\tilde{x}_{EWG}$ is $\mu$-GDP and we have,*

$$\mathbb{E}\, d(\tilde{x}_{EWG}, \bar{x}) \leq \sigma\sqrt{\frac{\pi}{2}} L_{1/2}^{d/2-1}\left(-\frac{d^2(p_0, f(\mathcal{D}))}{2}\right) + d(p_0, f(\mathcal{D})). \tag{3}$$

*where $L_{1/2}$ denote the Laguerre polynomials. With $p_0 = m_0$ and $\sigma \leq 2r/(n\mu)$, we have,*

$$\mathbb{E}\, d(\tilde{x}_{EWG}, \bar{x}) \leq \frac{2r}{n\mu}\sqrt{\frac{\pi}{2}} L_{1/2}^{d/2-1}\left(-\frac{r^2}{2}\right) + r.$$

*Proof.* See Appendix A.1.5. $\qquad\qquad\square$

Observe where the footpoint $p_0$ appears in the upper bounds (2) & (3). In the perfect scenario where the footpoint $p_0$ coincides with the true sample Fréchet mean $f(\mathcal{D})$, then the upper boundes reduces to $\sigma d$ and $\sigma\sqrt{\pi/2}$, respectively[3]. These upper bounds provide a perspective regarding how the selection of the footpoint affect the performance of the mechanisms. To shrink the upper bounds, we would like the quantity $d(p_0, f(\mathcal{D}))$ to be as small as possible. When the data is well-dispersed in $B_r(m_0)$, the true sample Fréchet mean will be near the center $m_0$. Therefore, the choice of $m_0$ as the footpoint is suitable. However, if there is prior knowledge that the ***majority*** of the data is clustered within some smaller region $R$ within $B_r(m_0)$, then the selection of the center of the region $R$ as the footpoint would be the better choice.

## 5 SIMULATIONS

In this section, we evaluate the utility of our Exponential-Wrapped Mechanisms, specifically in the context of releasing DP Fréchet means. Our numerical simulations are conducted on the space of symmetric positive definite matrices (SPDM), a manifold commonly used in medical imaging data (Pennec et al., 2019; Said et al., 2017; Hajri et al., 2016). Section 5.1 provides background information on the SPDM space. The setup and results of our simulations are detailed in Section 5.2. For further details on the simulations, please refer to Appendix A.2.

---

[3]Note that $L_{1/2}(0) = 1$ and $L_{1/2}(-x^2/2)$ is increasing as $x$ increases.

## 5.1 SPDM SPACE

First, we review some background material, refer to Hajri et al. (2016); Said et al. (2017); Reimherr et al. (2021) for more details. We denote the space of $m \times m$ real SPDM as $\mathcal{P}_m$ and identify it as a subspace of $\mathbb{R}^{m \times m}$ as $\mathcal{P}_m = \{A \in \mathbb{R}^{m \times m} : A^\top = A, x^\top A x > 0 \, \forall x \in \mathbb{R}^m, x \neq \mathbf{0}\}$. Similarly, at each $p \in \mathcal{P}_m$, we identify the tangent space $T_p \mathcal{P}_m$ as $\mathrm{Sym}(m)$, the space of $m \times m$ symmetric matrices. When equipping with the Rao-Fisher affine-invariant metric, $\mathcal{P}_m$ turns into a homogeneous Riemannian manifold of negative sectional curvature. Consider $X, Y \in T_p \mathcal{P}_m$, the Riemannian metric at $p$ is given by $\langle X, Y \rangle_p = \mathrm{tr}(p^{-1} X p^{-1} Y)$. It follows that the geodesic $\gamma$ connecting $p, q \in \mathcal{P}_m$ takes the form $\gamma(t) = p^{1/2} \left( p^{-1/2} q p^{-1/2} \right)^t p^{1/2}$. The distance function induced by the Riemannian metric turns out to be $d(p, q) = (\mathrm{tr}[\log_M(p^{-1/2} q p^{-1/2})^2])^{1/2}$ where $\log_M$ denotes the matrix logarithm. Under the Rao-Fisher metric, the Riemannian exponential and logarithm maps take the form $\mathrm{Exp}_p(X) = p^{1/2} \exp_M(p^{-1/2} X p^{-1/2}) p^{1/2}$ and $\mathrm{Log}_q(p) = q^{1/2} \log_M(q^{-1/2} p q^{-1/2}) q^{1/2}$.

## 5.2 SIMULATION RESULTS

First, we compare the performance between our Exponential-Wrapped Laplace mechanism described in Section 3.2 and the Riemannian Laplace mechanism proposed by Reimherr et al. (2021). We generate samples $\mathcal{D} = \{x_1, \ldots, x_n\}$ from a ball of radius $r$ centered at $\mathbf{I}_m$ on $\mathcal{P}_m$. The Fréchet mean $\bar{x}$ is computed using the gradient descent procedure described in Fletcher & Joshi (2004); Reimherr et al. (2021). To achieve $\varepsilon$-DP, the Frćhet mean $\bar{x}$ is perturbed using the Riemmanian Laplace mechanism and the Exponential-Wrapped Laplace mechanism. We employ the method from Hajri et al. (2016) with a burn-in iteration of $10,000$ to sample from the Riemannian Laplace mechanism and use the method described in Algorithm 1 with footprint $p_0 = \mathbf{I}_m$ to sample from the Exponential-Wrapped Laplace mechanism.

To evaluate the performance, we compute the Riemannian distance between the output of the mechanisms and the true Fréchet means. Denote $\tilde{x}_{\mathrm{RL}}^{dp}, \tilde{x}_{\mathrm{EWL}}^{dp}$ as the output of the Riemannian Laplace mechanism and Exponential-Wrapped Laplace mechanism, respectively.

Throughout these simulations, we fix the sample size at $n = 40$ to maintain a constant sensitivity $\Delta$. With $\Delta$ held constant, we varied the privacy budget $\varepsilon$ and manifold dimension $d = m(m+1)/2$. The top three plots in Figure 1 present the sample mean of the Riemannian distances $d(\bar{x}, \tilde{x}_{\mathrm{EWL}}^{dp})$ (depicted in red with circular symbols) and $d(\bar{x}, \tilde{x}_{\mathrm{RL}}^{dp})$ (in blue with triangular symbols) across 100 iterations, with the error band representing the sample mean $\pm$ standard error of the mean. Observing the plots, our Exponential-Wrapped Laplace mechanism yields comparable results to the Riemannian Laplace mechanism. However, as the dimension $d$ increases, its performance deteriorates relative to the Riemannian Laplace mechanism. This outcome is expected since the utility advantages of our exponential-wrapped Laplace mechanism over the Riemannian Laplace mechanism are reserved for non-homogeneous manifolds. Additionally, volume distortion becomes more significant as the dimension $d$ increases. The computational advantage of our mechanism is demonstrated in Table 1 as our mechanism is nearly 500 times faster to implement.

Next, we focus on releasing Fréchet mean in a GDP-compliant manner. We compare the performance between our Exponential-Wrapped Gaussian mechanism described in Section 3.2 and the Riemannian Laplace mechanism. Although the Riemannian Laplace mechanism is developed originally to achieve $\varepsilon$-DP, it's shown in Liu et al. (2022) any mechanism that satisfies $\varepsilon$-DP can achieve $\mu$-GDP with $\varepsilon = \log[(1 - \Phi(-u/2))/\Phi(-u/2)]$. Denote $\tilde{x}_{\mathrm{RL}}^{gdp}, \tilde{x}_{\mathrm{EWG}}^{gdp}$ as the output of the Riemannian Laplace mechanism and Exponential-Wrapped Laplace mechanism respectively, the bottom three plots in Figure 1 display the sample mean of the Riemannian distances $d(\bar{x}, \tilde{x}_{\mathrm{EWG}}^{gdp})$ (in red with circular symbols) and $d(\bar{x}, \tilde{x}_{\mathrm{RL}}^{gdp})$ (in blue with triangular symbols) across 100 iterations with the error band indicating the sample mean $\pm$ standard error. Across three dimensions $d \in \{3, 6, 15\}$, our Gaussian mechanism achieves better utility with smaller privacy budgets $\mu$. The Gaussian mechanism does not exhibit worse performance for $d = 15$ as in the case of the Exponential-Wrapped Laplace mechanism. The reason behind this is the quadratic decay of the Gaussian distribution does a better job of controlling the volume distortion.

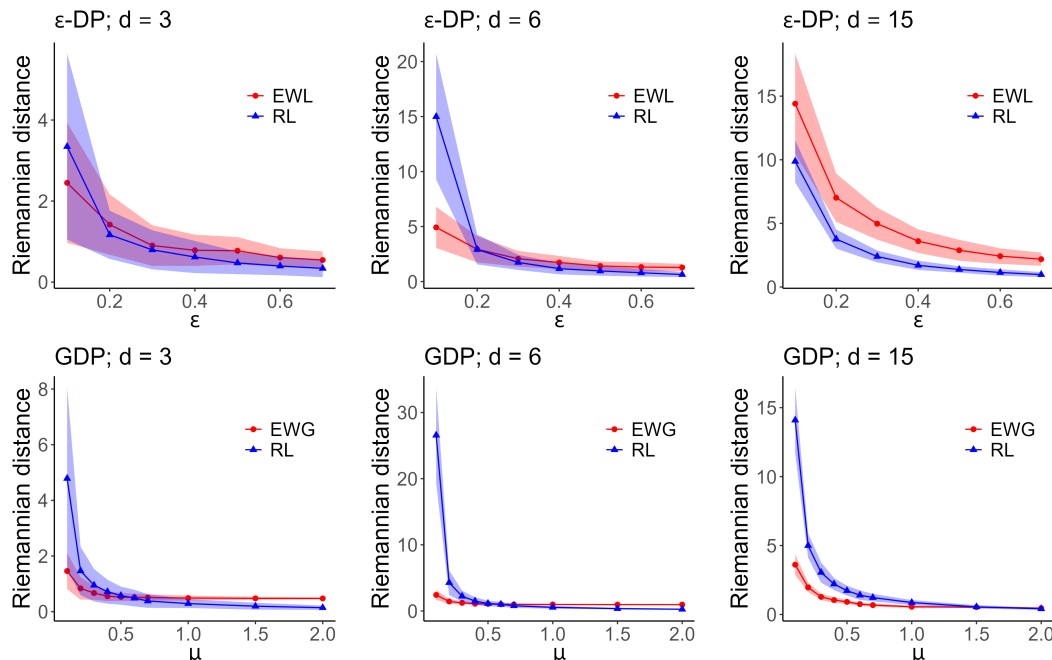

Figure 1: **Top three figures**: Simulation results for $\varepsilon$-DP. The blue lines with triangular symbols represent the sample mean of the Riemannian distances $d(\bar{x}, \tilde{x}_{\text{RL}}^{dp})$ across 100 repeats, while the red line with circular symbols indicates the Riemannian distances $d(\bar{x}, \tilde{x}_{\text{EWL}}^{dp})$. The error bands indicate the sample mean $\pm$ Standard Error. **Bottom three figures**: Simulation results for $\mu$-GDP. Blue lines with triangular symbols show the Riemannian distances $d(\bar{x}, \tilde{x}_{\text{RL}}^{gdp})$, and the red line with circular symbols represent the Riemannian distances $d(\bar{x}, \tilde{x}_{\text{EWG}}^{gdp})$. For more details, refer to Section 5.2. For additional experiment results, refer to Appendix A.3.

Lastly, we compare the computation times of the mechanisms as shown in Table 1. With a burn-in iteration of 10,000, the Riemannian Laplace mechanism takes an average of 1.54 seconds for $d = 15$. In contrast, our Laplace and Gaussian mechanisms take $4.06 \times 10^{-3}$ and $3.78 \times 10^{-3}$, respectively. Additionally, sampling from the Riemannian Laplace mechanism necessitates careful consideration of the proposal distribution during the MCMC procedure. During the simulation, we must fine-tune the rate of the proposal distribution to achieve reasonable results for the Riemannian Laplace mechanism with small $\mu$. Despite this, it can still produce unstable results, as observed in the two center plots for $\varepsilon = 0.1$ and $\mu = 0.1$.

Table 1: Computation times (mean $\pm$ standard error) for the Riemannian Laplace, Exponential-Wrapped Laplace, and Gaussian mechanisms.

| $d$ | Riemannian Laplace | Exponential-Wrapped Laplace | Exponential-Wrapped Gaussian |
|---|---|---|---|
| 3 | $1.05 \pm 7.09 \times 10^{-2}$ | $3.61 \times 10^{-3} \pm 5.96 \times 10^{-4}$ | $3.47 \times 10^{-3} \pm 8.51 \times 10^{-4}$ |
| 6 | $1.15 \pm 6.90 \times 10^{-2}$ | $3.62 \times 10^{-3} \pm 4.91 \times 10^{-4}$ | $3.53 \times 10^{-3} \pm 5.74 \times 10^{-4}$ |
| 15 | $1.54 \pm 8.96 \times 10^{-2}$ | $4.06 \times 10^{-3} \pm 7.45 \times 10^{-4}$ | $3.78 \times 10^{-3} \pm 1.27 \times 10^{-3}$ |

## 6 CONCLUSIONS AND FUTURE DIRECTIONS

In this paper, we develop Exponential-Wrapped Laplace and Gaussian mechanisms and apply them to achieve $\varepsilon$-DP, $(\varepsilon, \delta)$-DP, RDP, and GDP on Hadamard manifolds, the class of simply connected complete manifolds with non-positive curvature. We provide simple and fast algorithms for these

mechanisms. Finally, through simulations, we demonstrate that our Exponential-Wrapped Gaussian mechanism has comparable results and better results for small privacy budgets compared to the Riemannian Laplace mechanism for achieving GDP while having a magnitude of degree faster computational time.

For future research, the initial step should be to identify the optimal selection of the footpoint $p_0$ in both of our mechanisms. In non-constant curvature spaces, certain choices of $p_0$ may more effectively reduce the volume distortion introduced by the Riemannian exponential map. Since this paper focuses on manifolds of non-positive curvature, another direction would be to extend various differential privacy mechanisms to manifolds with non-negative curvature. Finally, instead of releasing privatized Fréchet means, we aim to extend our work to more complex tasks such as principal geodesic analysis (Huckemann et al., 2010; Fletcher et al., 2003; Zhang & Fletcher, 2013) and regression on manifolds (Cheng & Wu, 2013).

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

# A APPENDIX

## A.1 PROOFS

### A.1.1 PROOF OF THEOREM 3.1

*Proof.* Denote the Exponential-Wrapped Laplace mechanism as $M$ and its density as $g_1$ corresponding to $f(\mathcal{D})$ and $g_2$ corresponding to $f(\mathcal{D}')$. To show $\mathbb{P}(M(\mathcal{D}) \in S) \leq e^\varepsilon \mathbb{P}(M(\mathcal{D}') \in S)$ for all measurable set $S$, it's sufficient to show that,

$$\frac{g_1(y)}{g_2(y)} \leq e^\varepsilon.$$

We simplify the ratio on the left-hand side,

$$\frac{g_1(y)}{g_2(y)} = \frac{\frac{1}{J_{P_0}(\mathrm{Log}_{p_0}(y))} \exp\left(-\frac{\|\mathrm{Log}_{p_0}(y) - \mathrm{Log}_{p_0}(\eta_1)\|}{\sigma}\right)}{\frac{1}{J_{P_0}(\mathrm{Log}_{p_0}(y))} \exp\left(-\frac{\|\mathrm{Log}_{p_0}(y) - \mathrm{Log}_{p_0}(\eta_2)\|}{\sigma}\right)}$$

$$= \exp\left\{\frac{1}{\sigma}\left[\|\mathrm{Log}_{p_0}(y) - \mathrm{Log}_{p_0}(\eta_2)\| - \|\mathrm{Log}_{p_0}(y) - \mathrm{Log}_{p_0}(\eta_1)\|\right]\right\}$$

$$\leq \exp\left\{\frac{1}{\sigma}\|\mathrm{Log}_{p_0}(\eta_1) - \mathrm{Log}_{p_0}(\eta_2)\|\right\}, \quad \text{triangle inequality}$$

$$\leq \exp\left\{\frac{1}{\sigma}d(\eta_1, \eta_2)\right\}, \quad \log_y \text{ is a contraction for Hadamard manifold}$$

$$\leq \exp\left\{\frac{\Delta}{\sigma}\right\}$$

$$\leq e^\varepsilon, \quad \text{for } \sigma = \frac{\Delta}{\varepsilon}$$

$\square$

### A.1.2 PROOF OF THEOREM 3.2

*Proof.* Let $g_{p_0,\eta,\sigma}$ denote the Exponential-Wrapped Gaussian Distribution with footprint $p_0$, tangent center $\text{Log}_{p_0}(\eta)$ and rate $\sigma$. From Balle & Wang (2018), our Exponential-Wrapped Gaussain mechanism satisfies $(\varepsilon, \delta)$-DP if and only if,

$$\sup_{\mathcal{D} \simeq \mathcal{D}'} \int_A g_{p_0,\eta_1,\sigma}(y) \, d\nu(y) - e^\varepsilon \int_A g_{p_0,\eta_2,\sigma}(y) \, d\nu(y) \leq \delta$$

where $A = \{y \mid g_{p_0,\eta_1,\sigma}(y)/g_{p_0,\eta_2,\sigma}(y) \geq e^\varepsilon\}$, $\eta_1 = f(\mathcal{D})$ and $\eta_2 = f(\mathcal{D}')$. We have

$$\frac{g_{p_0,\eta_1,\sigma}(y)}{g_{p_0,\eta_2,\sigma}(y)}$$

$$= \exp\left\{\frac{1}{2\sigma^2}\left[\|\text{Log}_{p_0}(y) - \text{Log}_{p_0}(\eta_2)\|^2 - \|\text{Log}_{p_0}(y) - \text{Log}_{p_0}(\eta_1)\|^2\right]\right\}$$

$$= \exp\left\{\frac{1}{2\sigma^2}\left[-2\left\langle\text{Log}_{p_0}(y) - \text{Log}_{p_0}(\eta_1), \text{Log}_{p_0}(\eta_2) - \text{Log}_{p_0}(\eta_1)\right\rangle + \|\text{Log}_{p_0}(\eta_2) - \text{Log}_{p_0}(\eta_1)\|^2\right]\right\}$$

Denote $\Delta_{p_0,\eta_1,\eta_2} = \|\text{Log}_{p_0}(\eta_2) - \text{Log}_{p_0}(\eta_1)\|$. It follows that,

$$A = \left\{y \mid \left\langle\text{Log}_{p_0}(y) - \text{Log}_{p_0}(\eta_1), \text{Log}_{p_0}(\eta_2) - \text{Log}_{p_0}(\eta_1)\right\rangle \leq -\sigma^2\varepsilon + \frac{\Delta_{p_0,\eta_1,\eta_2}^2}{2}\right\}$$

Apply change of variable with $u = \text{Log}_{p_0} y$, we have

$$\sup_{\mathcal{D} \simeq \mathcal{D}'} \int_{A^*} \mathcal{N}(u \mid \text{Log}_{p_0}(\eta_1), \sigma^2\mathbf{I}) \, d\lambda(u) - e^\varepsilon \int_{A^*} \mathcal{N}(u \mid \text{Log}_{p_0}(\eta_2), \sigma^2\mathbf{I}) \, d\lambda(u) \leq \delta$$

where $\lambda$ is the Lebegue measure on the tangent space $T_{p_0}\mathcal{M}$ and

$$A^* = \left\{u \mid \left\langle u - \text{Log}_{p_0}(\eta_1), \text{Log}_{p_0}(\eta_2) - \text{Log}_{p_0}(\eta_1)\right\rangle \geq -\sigma^2\varepsilon + \frac{\Delta_{p_0,\eta_1,\eta_2}^2}{2}\right\}$$

It follows that,

$$\int_{A^*} \mathcal{N}(u \mid \text{Log}_{p_0}(\eta_1), \sigma^2\mathbf{I}) \, d\lambda(u) = \Phi\left(-\frac{\sigma\varepsilon}{\Delta_{p_0,\eta_1,\eta_2}} + \frac{\Delta_{p_0,\eta_1,\eta_2}}{2\varepsilon}\right).$$

Take a similar approach for the second integral, we have

$$\int_{A^*} \mathcal{N}(u \mid \text{Log}_{p_0}(\eta_2), \sigma^2\mathbf{I}) \, d\lambda(u) = \Phi\left(-\frac{\sigma\varepsilon}{\Delta_{p_0,\eta_1,\eta_2}} - \frac{\Delta_{p_0,\eta_1,\eta_2}}{2\varepsilon}\right).$$

Finally, we have

$$\Phi\left(-\frac{\sigma\varepsilon}{\Delta_{p_0}} + \frac{\Delta_{p_0}}{2\sigma}\right) - e^\varepsilon\Phi\left(-\frac{\sigma\varepsilon}{\Delta_{p_0}} + \frac{\Delta_{p_0}}{2\sigma}\right) \leq \delta$$

where $\Delta_{p_0} = \sup_{\mathcal{D} \simeq \mathcal{D}'} \Delta_{p_0,\eta_1,\eta_2}$ as needed. $\qquad\square$

### A.1.3 PROOF OF THEOREM 3.3

*Proof.* Using definition 2.3, we need to show the following,

$$\forall \varepsilon \geq 0, \ \sup_{\mathcal{D} \simeq \mathcal{D}'} \int_A g_{p_0,\eta_1,\sigma}(y) \, d\nu(y) - e^\varepsilon \int_A g_{p_0,\eta_2,\sigma}(y) \, d\nu(y) \leq \delta_\mu(\varepsilon) \tag{4}$$

where $g$ denotes the density of the Exponential-Wrapped Gaussian Distribution. From the proof in A.1.2, we have

$$\sup_{\mathcal{D} \simeq \mathcal{D}'} \int_A g_{p_0,\eta_1,\sigma}(y) \, d\nu(y) - e^\varepsilon \int_A g_{p_0,\eta_2,\sigma}(y) \, d\nu(y)$$

$$= \Phi\left(-\frac{\sigma\varepsilon}{\Delta_{p_0}} + \frac{\Delta_{p_0}}{2\sigma}\right) - e^\varepsilon\Phi\left(-\frac{\sigma\varepsilon}{\Delta_{p_0}} + \frac{\Delta_{p_0}}{2\sigma}\right).$$

Thus, the equality in (4) holds if and only if $\sigma = \Delta_{p_0}/\mu$. Since $\text{Log}_{p_0}$ is a contraction for any $p_0 \in \mathcal{M}$ (for Hadamard manifold $\mathcal{M}$), we have $\Delta \geq \Delta_{p_0}$ and $\sigma = \Delta/\mu$ achieves $\mu$-GDP as well.

$\qquad\square$

### A.1.4 PROOF OF THEOREM 3.4

*Proof.* Let $M$ denote the Exponential-Wrapped Gaussian mechanism, we have

$$D_\alpha(M(\mathcal{D})\|M(\mathcal{D}'))$$

$$= \frac{1}{\alpha - 1} \log \int \frac{1}{J_{p_0}(\mathrm{Log}_{p_0}(y))} \frac{1}{(\sqrt{2\pi}\sigma)^d}$$

$$\exp\left\{-\frac{\alpha}{2\sigma^2}\left[\|\mathrm{Log}_{p_0} y - \mathrm{Log}_{p_0}\eta_1\|^2\right] - \frac{1-\alpha}{2\sigma^2}\|\mathrm{Log}_{p_0} y - \mathrm{Log}_{p_0}\eta_2\|^2\right\} d\nu(y)$$

$$= \frac{1}{\alpha - 1} \log \exp\left\{-\frac{\alpha(1-\alpha)}{2\sigma^2}\|\mathrm{Log}_{p_0}\eta_1 - \mathrm{Log}_{p_0}\eta_2\|^2\right\}, \text{ completing the squares}$$

$$= \frac{\alpha}{2\sigma^2}\|\mathrm{Log}_{p_0}\eta_1 - \mathrm{Log}_{p_0}\eta_2\|^2$$

$$\leq \frac{\alpha}{2\sigma^2}d(\eta_1 - \eta_2)^2, \ \mathrm{Log}_{p_0} \text{ is a contraction for Hadamard manifolds}$$

$$\leq \frac{\alpha}{2\sigma^2}\Delta^2$$

$$\leq \varepsilon, \text{ for } \sigma = \frac{\Delta}{\sqrt{2\varepsilon/\alpha}}$$

$\square$

### A.1.5 PROOF OF THEOREM 4.1

**Lemma A.0.1.** *Let $\mathcal{M}$ be a $d$-dimensional Hadamard manifold,*

1. *denote $y$ as a sample drawn from an Exponential-Wrapped Laplace Distribution with footprint $p_0$, tangent center $\mathrm{Log}_{p_0}\eta$ and rate $\sigma$, then we have,*

$$\mathbb{E}\, d(y, \eta) \leq \sigma d + 2\|\mathrm{Log}_{p_0}\eta\|$$

2. *denote $y$ as a sample drawn from an Exponential-Wrapped Gaussian Distribution with footprint $p_0$, tangent center $\mathrm{Log}_{p_0}\eta$ and rate $\sigma$, then we have*

$$\mathbb{E}\, d(y, \eta) \leq \sigma\sqrt{\frac{\pi}{2}} L_{1/2}^{d/2-1}\left(-\frac{d(p_0,\eta)^2}{2}\right) + d(p_0,\eta) \leq \sigma\sqrt{2}\frac{\Gamma((d+1)/2)}{\Gamma(d/2)} + 2\|\mathrm{Log}_{p_0}\eta\|.$$

*Proof.* For Exponential-Wrapped Laplace Distribution, denote

$$C(\sigma) = \int \exp\left(-\frac{\|x\|}{\sigma}\right) d\lambda(x),$$

then we have

$$\mathbb{E}d(y,\eta)$$

$$= \int d(y,\eta)\frac{C(\sigma)^{-1}}{J_{p_0}(\mathrm{Log}_{p_0} y)} \exp\left(-\frac{\|\mathrm{Log}_{p_0} y - \mathrm{Log}_{p_0}\eta\|}{\sigma}\right) d\nu(y)$$

$$\leq \int d(y,p_0)\frac{C(\sigma)^{-1}}{J_{p_0}(\mathrm{Log}_{p_0} y)} \exp\left(-\frac{\|\mathrm{Log}_{p_0} y - \mathrm{Log}_{p_0}\eta\|}{\sigma}\right) d\nu(y) + d(p_0,\eta), \text{ triangle inequality}$$

$$= \int \|\mathrm{Log}_{p_0} y\|\frac{C(\sigma)^{-1}}{J_{p_0}(\mathrm{Log}_{p_0} y)} \exp\left(-\frac{\|\mathrm{Log}_{p_0} y - \mathrm{Log}_{p_0}\eta\|}{\sigma}\right) d\nu(y) + d(p_0,\eta)$$

$$= \int \frac{\|u + \mathrm{Log}_{p_0}\eta\|}{C(\sigma)} \exp\left(-\frac{\|u\|}{\sigma}\right) d\lambda(u) + d(p_0,\eta), \ u = \mathrm{Log}_{p_0} y - \mathrm{Log}_{p_0}\eta$$

$$\leq \frac{1}{C(\sigma)} \int \|u\| \exp\left(-\frac{\|u\|}{\sigma}\right) d\lambda(u) + 2d(p_0,\eta), \text{ triangle inequality}$$

$$= \left(\sigma \int_0^\infty r^{d-1}\exp(-r)\, dr\right)^{-1} \int_0^\infty \sigma^2 r^d \exp(-r)\, dr + 2d(p_0,\eta), \text{ spherical coordinates}$$

$$= \sigma d + 2d(p_0,\eta)$$

The lower bound can be derived similarly and using the fact that $\|\operatorname{Log}_{p_0} y - \operatorname{Log}_{p_0} \eta\| \leq d(y, \eta)$. Similarly, for exponential-wrapped Gaussian distribution, we have,

$$
\mathbb{E}d(y, \eta)
$$

$$
= \int d(y, \eta) \frac{(\sqrt{2\pi}\sigma)^{-d}}{J_{p_0}(\operatorname{Log}_{p_0} y)} \exp\left(-\frac{\|\operatorname{Log}_{p_0} y - \operatorname{Log}_{p_0} \eta\|^2}{2\sigma^2}\right) d\nu(y)
$$

$$
\leq \mathbb{E}d(y, p_0) + d(p_0, \eta)
$$

Note that since $(\operatorname{Log}_{p_0} y)/\sigma \sim \mathcal{N}(\operatorname{Log}_{p_0} \eta, \mathbf{I})$, $d(y, p_0)/\sigma$ follows a noncentral chi distribution and have a mean of

$$
\sqrt{\frac{\pi}{2}} L_{1/2}^{d/2-1}\left(-\frac{d(p_0, \eta)^2}{2}\right)
$$

where $L_{1/2}$ denote the Laguerre polynomials. Thus, we have

$$
\mathbb{E}d(y, \eta)
$$

$$
\leq \sigma\sqrt{\frac{\pi}{2}} L_{1/2}^{d/2-1}\left(-\frac{d(p_0, \eta)^2}{2}\right) + d(p_0, \eta)
$$

However, this upper bound is hard to interpret. We will also derive a less tight upper bound but with better interpretability as follows.

$$
\mathbb{E}d(y, \eta)
$$

$$
= \int d(y, \eta) \frac{(\sqrt{2\pi}\sigma)^{-d}}{J_{p_0}(\operatorname{Log}_{p_0} y)} \exp\left(-\frac{\|\operatorname{Log}_{p_0} y - \operatorname{Log}_{p_0} \eta\|^2}{2\sigma^2}\right) d\nu(y)
$$

$$
\leq \int \frac{\|u + \operatorname{Log}_{p_0} \eta\|}{(\sqrt{2\pi}\sigma)^d} \exp\left(-\frac{\|u\|^2}{2\sigma}\right) d\lambda(u) + d(p_0, \eta), \ u = \operatorname{Log}_{p_0} y - \operatorname{Log}_{p_0} \eta
$$

$$
\leq \int \frac{\|u\|}{(\sqrt{2\pi}\sigma)^d} \exp\left(-\frac{\|u\|^2}{2\sigma}\right) d\lambda(u) + 2d(p_0, \eta), \ \text{triangle inequality}
$$

$$
= \sigma\sqrt{2} \frac{\Gamma((d+1)/2)}{\Gamma(d/2)} + 2d(p_0, \eta), \ \text{since } \frac{\|u\|}{\sigma} \sim \chi_d
$$

The lower bound can be derived similarly and using the fact that $\|\operatorname{Log}_{p_0} y - \operatorname{Log}_{p_0} \eta\| \leq d(y, \eta)$. $\qquad \square$

Theorem 4.1 follows from Lemma A.0.1 directly.

## A.2 R CODES

For simulations in section 5, refer to simulation_laplace.R for simulation on $\varepsilon$-DP and simulation_gaussian.R for simulation on GDP. vanilla_DP_plot.R and GDP_plot.R are for generating the result plots in Figure 1.

The simulations were performed using R on a PC with a 12th Gen Intel Core i5-12600K CPU with 32 GB of RAM running Windows 11.

## A.3 ADDITIONAL EXPERIMENT RESULTS

### A.3.1 ADDITIONAL EXPERIMENT RESULTS ON REAL-WORLD DATASET

Here, we provide additional experiment results on the real-world dataset, OCTMNIST, from the biomedical datasets MedMNISTS in Yang et al. (2023). OCTMNIST consists of $28 \times 28$ greyscale images.

The MedMNIST dataset includes 12 standardized 2D datasets and 6 standardized 3D datasets, sourced from carefully curated medical imaging modalities such as X-ray, OCT, ultrasound, CT, and electron microscopy. It supports a variety of classification tasks, including binary, multi-class, ordinal regression, and multi-label classification, with dataset sizes ranging from 100 to 100,000 samples.

To obtain the covariance descriptor from the data, we follow a similar approach as in Utpala et al. (2023a); Tuzel et al. (2006). Let $\mathcal{I} \in \mathbb{R}^{h \times w}$ be an greyscale image of height $h$ and width $w$. The covariance descriptor is given by

$$R_\eta(\mathcal{I}) = \left[ \frac{1}{|\mathcal{S}|} \sum_{\mathbf{x} \in S} (\phi(\mathcal{I})(\mathbf{x}) - \mu)(\phi(\mathcal{I})(\mathbf{x}) - \mu)^T \right] + \eta \mathbf{I},$$

where

$$\phi(\mathcal{I})(\mathbf{x}) = \left[ \mathcal{I}(x, y), \left| \frac{\partial \mathcal{I}(x, y)}{\partial x} \right|, \left| \frac{\partial \mathcal{I}(x, y)}{\partial y} \right|, \left| \frac{\partial^2 \mathcal{I}(x, y)}{\partial x^2} \right|, \left| \frac{\partial^2 \mathcal{I}(x, y)}{\partial y^2} \right| \right],$$

and we set $\eta = 10^{-6}$.

Follow a similar computation as in Utpala et al. (2023a), we have

$$d(R_\eta(\mathcal{I}), \mathbf{I}) \leq \sqrt{5} \max \left\{ |\log(\eta)|, |\log(5 \cdot 255^2 + \eta)| \right\} \approx 31. \tag{5}$$

Note that different from the experiment in Utpala et al. (2023a), we did not normalize the pixel value/intensity $\mathcal{I}$ to be between $0$ and $1$. Based on (5), the data must reside in $B_r(\mathbf{I})$ where $r$ is the righthand side of (5) and thus the sensitivity is $\Delta = \sup_{\bar{x} \simeq \bar{x}'} d(\bar{x}, \bar{x}') \leq 2r/n$.

Similarly to the simulation study in section 5, we focus on outputting the Fréchet mean in a GDP-compliant way. Since the radius of the geodesic ball the data resides in is fairly large and we don't have any prior information on the data distribution, it's necessary to use part of the privacy budget to select a good footpoint. Given the total privacy budget $\mu$, we use $\sqrt{0.1}\mu$ to select the footpoint and $\sqrt{0.9}\mu$ to output the Fréchet mean. To select the footpoint, we uniformly sample $5\%$ of data, compute its Fréchet mean and privatize it using the Riemannian Laplace mechanism in Reimherr et al. (2021).

There are four different classes in the OCTMNIST dataset, labelled from $0$ to $3$. Denote $\tilde{x}_{\text{RL}}^{gdp}, \tilde{x}_{\text{EWG}}^{gdp}$ as the output of the Riemannian Laplace mechanism and Exponential-Wrapped Laplace mechanism respectively, the two plots in Figure 2 display the sample mean of the Riemannian distances $d(\bar{x}, \bar{x}_{\text{EWG}}^{gdp})$ (in red with circular symbols) and $d(\bar{x}, \bar{x}_{\text{RL}}^{gdp})$ (in blue with triangular symbols) across $10$ iterations for classes labelled $1$ and $2$. Similarly, Figure 3 shows the result for classes labelled $0$ and $3$.

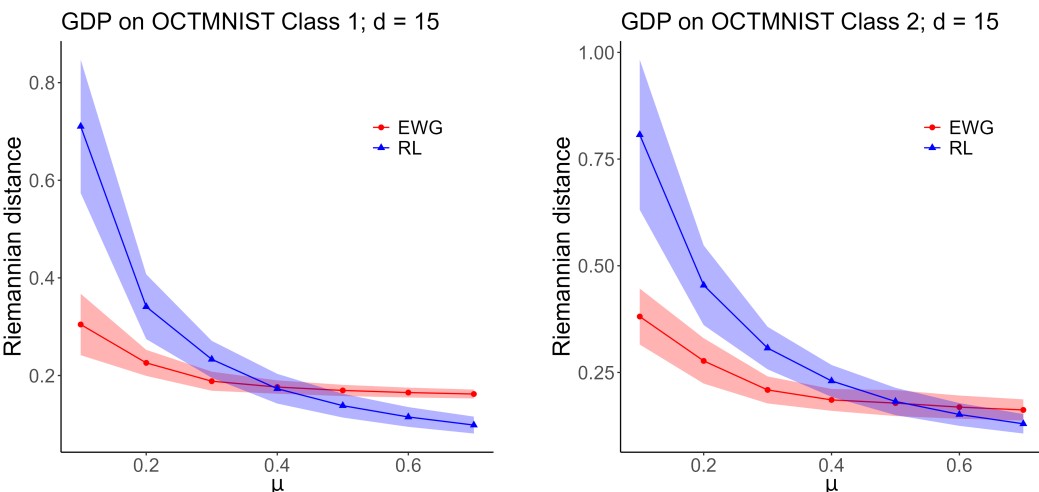

Figure 2: Results for Class 1 and Class 2 in the OCTMNIST data.

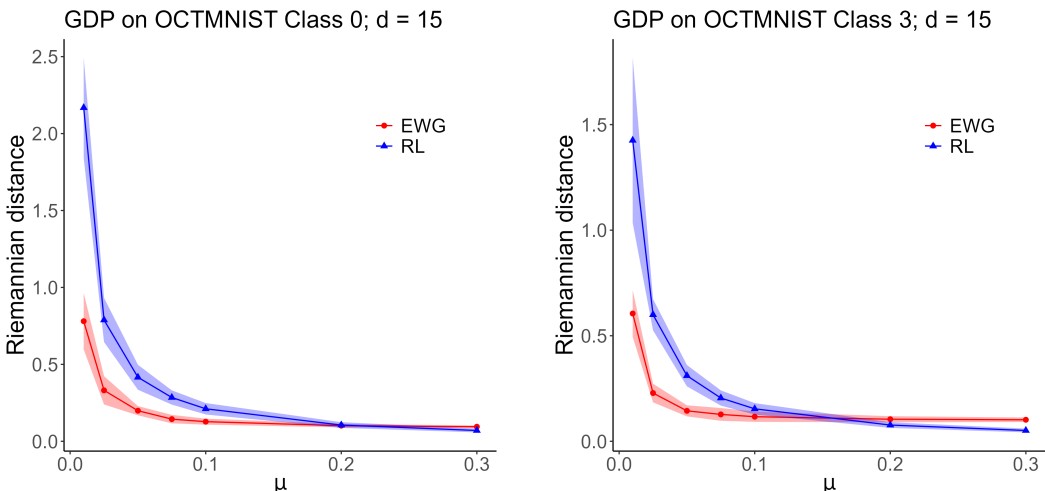

Figure 3: Results for Class 0 and Class 3 in the OCTMNIST data.

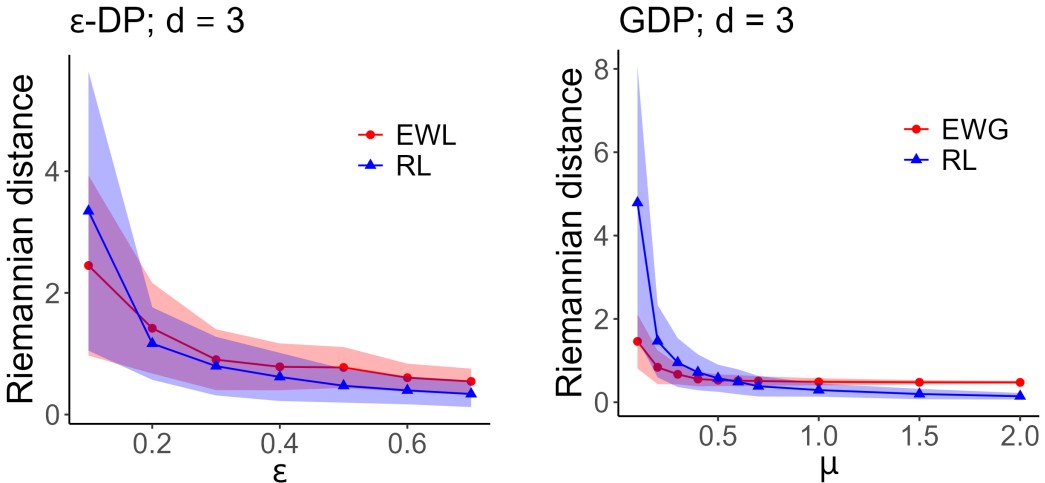

Figure 4: Simulation results for $\varepsilon$-DP and $\mu$-GDP with $d = 3$.

### A.3.2 ADDITIONAL SIMULATION RESULTS

### A.4 BROADER IMPACTS

Differential privacy on manifolds ensures robust privacy protections for datasets with complex geometric structures, such as those found in healthcare, geography, and neuroscience. By tailoring privacy mechanisms to the unique properties of manifolds, we reduce re-identification risks, improve data utility and promote ethical data use. This approach builds public trust, supports open data initiatives, and drives innovation by enabling secure analysis of manifold-based data.

The challenge with differential privacy on manifolds lies in grasping and applying the manifold concept itself. Adapting privacy mechanisms to these intricate geometric structures requires a deep understanding, which might not be readily accessible, particularly for smaller organizations. This complexity could slow down the adoption of these methods, potentially delaying progress in fields such as healthcare, geospatial analysis, and scientific research, where manifold-based data holds significant potential.

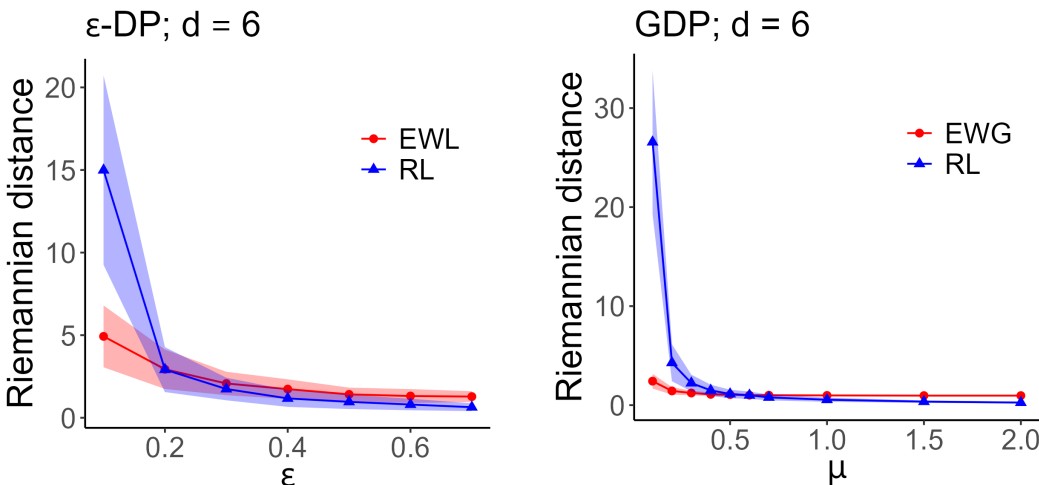

Figure 5: Simulation results for $\varepsilon$-DP and $\mu$-GDP with $d = 6$.

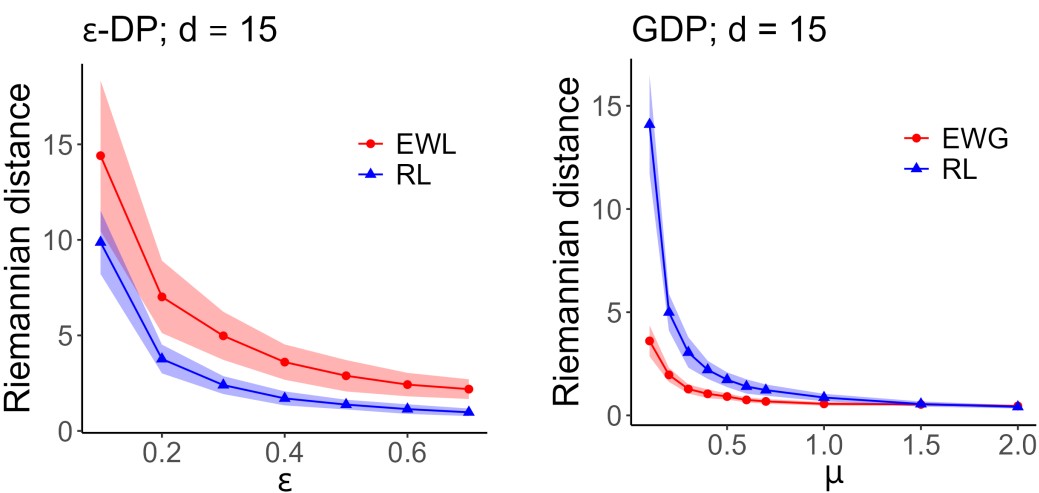

Figure 6: Simulation results for $\varepsilon$-DP and $\mu$-GDP with $d = 15$.

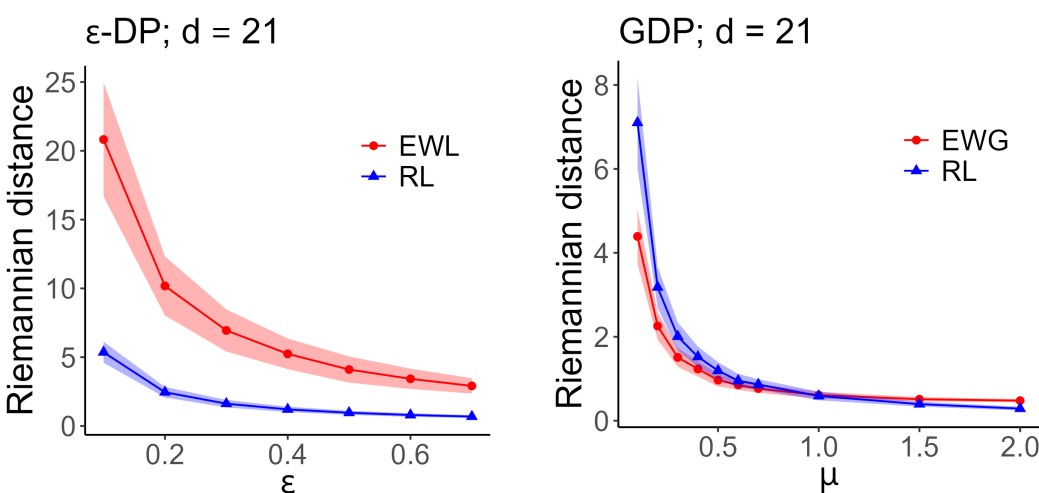

Figure 7: Simulation results for $\varepsilon$-DP and $\mu$-GDP with $d = 21$.

