# OpenReview forum: "Exponential-Wrapped Mechanisms for Differential Privacy on Hadamard Manifolds"
_ICLR.cc/2025/Conference — Submitted to ICLR 2025_

### Official Review · Reviewer_rzys · 2024-10-28

**Soundness:** 3
**Presentation:** 2
**Contribution:** 3
**Rating:** 6
**Confidence:** 4

**Summary:**

This paper extends the framework of Differential Privacy (DP) to Hadamard manifolds, a class of complete and simply connected Riemannian manifolds with non-positive sectional curvature. Inspired by the Cartan–Hadamard theorem, the authors introduce Exponential-Wrapped Laplace and Gaussian mechanisms to achieve ε-DP, (ε, δ)-DP, Gaussian DP (GDP), and Rényi DP (RDP) on these manifolds. Unlike traditional methods that rely on computationally intensive Monte Carlo Markov Chain (MCMC) techniques, their approach uses efficient algorithms that bypass these processes. This work is claimed to be the first to extend (ε, δ)-DP, GDP, and RDP to Hadamard manifolds.

The effectiveness of their method is demonstrated through simulations on the space of Symmetric Positive Definite Matrices (SPDM), a frequently used Hadamard manifold in statistics. The results show that the proposed Exponential-Wrapped mechanisms surpass traditional MCMC-based methods in both performance and ease of implementation, especially in scenarios requiring small privacy budgets (ε). Moreover, their approach achieves comparable utility to the Riemannian Laplace mechanism while operating at significantly faster computational speeds.

The paper concludes by suggesting possible future research directions, including extending differential privacy to manifolds with non-negative curvature and applying the methods to more complex tasks such as principal geodesic analysis and regression on manifolds.

**Strengths:**

The paper is the first to extend $(\epsilon, \delta)$-DP, GDP, and RDP to Hadamard Riemannian manifolds through the introduction of Exponential-Wrapped Laplace and Gaussian mechanisms. This extension broadens the scope of privacy frameworks to general manifold settings.

The authors develop fast and efficient implementations of these mechanisms, avoiding computationally intensive MCMC sampling methods. This makes the proposed methods more feasible and applicable in real-world scenarios where computational efficiency is crucial.

Through comprehensive numerical experiments, the paper demonstrates that the proposed mechanisms perform comparably to the traditional Riemannian Laplace mechanism. Notably, when achieving GDP, the Exponential-Wrapped Gaussian mechanism exhibits superior performance, especially in scenarios with small privacy budgets.

**Weaknesses:**

The paper lacks comparison with previous results. There is no clear comparison within the sections to evaluate the proposed method against existing methods, making it difficult for readers to assess whether the proposed approach truly represents an improvement. Additionally, too many parameters and concepts are introduced, with overly exotic ideas seemingly added to elevate the paper's complexity. The authors may have introduced unnecessary complexity to attract readers’ attention. The practical applications are limited, only valid on Hadamard manifolds. This does not differ significantly from cases on $\mathbb{R}^n$ but has greater limitations. After all, when do people operate in a space of negative curvature?

**Questions:**

Could the authors provide specific examples? Also, what is the motivation for considering Hadamard manifolds in this paper? What considerations led the authors to naturally introduce Hadamard manifolds into differential privacy? Additionally, is this related to the stronger divergence of the Laplace mechanism and the better behavior of its solution, such as better convergence of its inverse? If you can address the above questions, I will consider adjusting the score accordingly.

---

> ### Author Response · Authors · 2024-11-19
> **Reply to Weakness**
>
> Thank you for your thorough review and constructive feedback. We appreciate your insights, and we have addressed your concerns and questions in our response below:
>
> ## Weakness
>
> > The paper lacks comparison with previous results. There is no clear comparison within the sections to evaluate the proposed method against existing methods, making it difficult for readers to assess whether the proposed approach truly represents an improvement.
>
>
> For our Exponential-Wrapped (EW) Laplace mechanism, we conducted a comparison with the Riemannian Laplace mechanism proposed in [1] on SPDM spaces. Our results show that the EW Laplace mechanism achieves comparable performance overall. As the dimension $d$ increases, its performance declines relative to the Riemannian Laplace mechanism, which is expected since the utility benefits of our mechanism are designed for non-homogeneous manifolds. Specifically, the EW Laplace mechanism uses a sensitivity parameter of $\Delta/\varepsilon$ instead of the $2\Delta/\varepsilon$ used by the Riemannian mechanism. However, this trade-off leads to a substantial computational advantage: the EW Laplace mechanism is nearly 400 times faster. This makes it highly practical for applications requiring computational efficiency.
>
>
> 1. Performance Metrics: Please see the top three plots in Figure 1.
> 2. Computational Time Comparison: Refer to the first two columns of Table 1.
>
> For our Exponential-Wrapped (EW) Gaussian mechanism, we focused on achieving GDP on SPDM spaces. To our knowledge, this is the first mechanism to achieve GDP (or any variant of DP definition other than pure-DP) on SPDM spaces with commonly used metrics, such as the affine-invariant metric. Existing works are limited: [2] supports GDP only on constant curvature spaces, and [3] achieves $(\varepsilon,\delta)$-DP for SPDMs but is restricted to the log-Euclidean metric. Our EW Gaussian mechanism generalizes [3], working for any Riemannian metric, and reduces to the tangent Gaussian mechanism under the log-Euclidean metric.
>
> In the absence of comparable GDP mechanisms for SPDM spaces, we evaluated the EW Gaussian mechanism against the Riemannian Laplace mechanism [1]. Our Gaussian mechanism demonstrates superior utility at smaller privacy budgets $\mu$.
>
>
> 1. Performance Metrics: Please see the bottom three plots in Figure 1.
> 2. Computational Time Comparison: Refer to the last column of Table 1.
>
>
>
> > The practical applications are limited, only valid on Hadamard manifolds. This does not differ significantly from cases on $\mathbb{R}^n$ but has greater limitations. After all, when do people operate in a space of negative curvature?
>
>
> We believe there are many practical applications in Hadamard manifolds. Hadamard manifolds contain two of the most encountered manifolds in statistics/machine learning, the hyperbolic space and the space of SPD matrices.
>
>
> SPD matrices play a significant role in medical imaging and are often called tensors when modelling the covariance matrix of water's Brownian motion in Diffusion Tensor Imaging [4, 5]. They have also been used in shape analysis to capture joint variability across different locations [6, 7]. Furthermore, SPD matrices are widely applied in computer vision and image analysis tasks, including segmentation, grouping, and motion analysis [8, 9, 10, 11]. For a summary of the practical applications of SPDM space, refer to Chapter 3 of [12].
>
> Hyperbolic spaces have gained considerable attention in machine learning for their ability to enhance performance and efficiency, particularly in learning representations of hierarchical structures and graph data [13, 14]. A key characteristic of hyperbolic space is its exponential volume growth, which allows tree-like hierarchical data to be embedded with low distortion while requiring only a few dimensions [15]. It leads to lower complexity, fewer parameters, and less overfitting. For more details, refer to [16].

---

> ### Author Response · Authors · 2024-11-19
> **Reply to Questions**
>
> ## Question
>
>
> > Could the authors provide specific examples? Also, what is the motivation for considering Hadamard manifolds in this paper? What considerations led the authors to naturally introduce Hadamard manifolds into differential privacy?
>
> The key motivation behind considering Hadamard manifolds centers around the space of SPDM. As mentioned, it's one of the most important manifolds in statistics and machine learning. In particular, it's commonly used in biomedical data such as brain imaging. Brain imaging data, such as MRI and fMRI scans, often contain highly detailed and unique features that could inadvertently reveal identities or medical conditions. Developing differential privacy mechanisms on Hadamard manifolds (and thus SPDM space) allows us to protect sensitive patient information while enabling valuable research and analysis.
>
>
>
> > Additionally, is this related to the stronger divergence of the Laplace mechanism and the better behavior of its solution, such as better convergence of its inverse?
>
> Can you provide a bit more explanation on this particular comment? We are a little unsure on what are the meanings of "divergence of the Laplace mechanism" and "better convergence of its inverse".
>
>
> ## References
>
> [1]: Matthew Reimherr, Karthik Bharath, and Carlos Soto. Differential privacy over Riemannian manifolds. In Advances in Neural Information Processing Systems, volume 34, pp. 12292–12303. Curran Associates, Inc., 2021.
>
> [2]: Yangdi Jiang, Xiaotian Chang, Yi Liu, Lei Ding, Linglong Kong, and Bei Jiang. Gaussian differential privacy on Riemannian manifolds. In Advances in Neural Information Processing Systems 36: Annual Conference on Neural Information Processing Systems 2023.
>
> [3]: Saiteja Utpala, Praneeth Vepakomma, and Nina Miolane. Differentially private fr\'echet mean on the manifold of symmetric positive definite (SPD) matrices with log-euclidean metric. Transactions on Machine Learning Research, 2023.
>
> [4]: Peter J. Basser, James Mattiello, Denis Le Bihan, MR diffusion tensor spectroscopy and imaging, Biophysical Journal 66 (1994) 259–267.
>
> [5]: Denis Le Bihan, Jean-François Mangin, Cyril Poupon, C.A. Clark, S. Pappata, N. Molko, H. Chabriat, Diffusion tensor imaging: concepts and applications, Journal of Magnetic Resonance Imaging 13 (4) (2001) 534–546.
>
> [6]: Pierre Fillard, Vincent Arsigny, Xavier Pennec, Paul M. Thompson, Nicholas Ayache, Extrapolation of sparse tensor fields: application to the modelling of brain variability, in: Gary Christensen, Milan Sonka (Eds.), Proc. of Information Processing in Medical Imaging 2005 (IPMI’05), Glenwood Springs, Colorado, USA, in: LNCS, vol. 3565, Springer, July 2005, pp. 27–38, PMID: 17354682.
>
> [7]: Pierre Fillard, Vincent Arsigny, Xavier Pennec, Kiralee M. Hayashi, Paul M. Thompson, Nicholas Ayache, Measuring brain variability by extrapolating sparse tensor fields measured on sulcal lines, NeuroImage 34 (2) (January 2007) 639–650, also as INRIA Research Report 5887, April 2006, PMID: 17113311.
>
> [8]: Gérard Medioni, Mi-Suen Lee, Chi-Keung Tang, A Computational Framework for Segmentation and Grouping, Elsevier, 2000.
>
> [9]: Thomas Brox, Johachim Weickert, Bernhard Burgeth, Pavel Mrázek, Nonlinear structure tensors, Image and Vision Computing 24 (1) (2006) 41–55.
>
> [10]: Johachim Weickert, Thomas Brox, Diffusion and regularization of vector- and matrix-valued images, in: M.Z. Nashed, O. Scherzer (Eds.), Inverse Problems, Image Analysis, and Medical Imaging, in: Contemporary Mathematics, vol. 313, AMS, Providence, 2002, pp. 251–268.
>
> [11]: Joachim Weickert, Hans Hagen (Eds.), Visualization and Processing of Tensor Fields. Mathematics and Visualization, Springer, 2006.
>
> [12]: Xavier Pennec, Stefan Sommer, and Tom Fletcher. Riemannian geometric statistics in medical image analysis. Academic Press, 2019.
>
> [13]: Maximillian Nickel and Douwe Kiela. Poincare embeddings for learning hierarchical representations. Advances in neural information processing systems, 30, 2017.
>
> [14]: Benjamin P Chamberlain, James Clough, and Marc P Deisenroth. Neural embeddings of graphs in hyperbolic space. arXiv preprint arXiv:1705.10359, 2017.
>
> [15]: Rik Sarkar. Low distortion delaunay embedding of trees in hyperbolic plane. In International Symposium on Graph Drawing, pp. 355–366. Springer, 2011.
>
> [16]: Edoardo Cetin, Benjamin Paul Chamberlain, Michael M. Bronstein, and Jonathan J Hunt. Hyperbolic deep reinforcement learning. In The Eleventh International Conference on Learning Representations, 2023.

---

> > ### Comment · Reviewer_rzys · 2024-11-26
> >
> > Thanks for your reply, I will adjust my score.

---

> > > ### Author Response · Authors · 2024-11-27
> > >
> > > Thank you for your feedback and for increasing the score. We appreciate your time and consideration.

---

### Official Review · Reviewer_JTT4 · 2024-11-02

**Soundness:** 3
**Presentation:** 3
**Contribution:** 4
**Rating:** 8
**Confidence:** 4

**Summary:**

This paper introduces novel mechanisms for achieving differential privacy on Hadamard manifolds using exponential-wrapped distributions. The authors extend multiple variations of differential privacy (DP) - including (ε,δ)-DP, Gaussian DP (GDP), and Rényi DP (RDP) - to Hadamard manifolds for the first time. The key innovation is using exponential-wrapped Laplace and Gaussian distributions to create mechanisms that are both theoretically sound and computationally efficient, avoiding the need for complex MCMC sampling methods used in previous approaches. The paper provides theoretical guarantees for privacy and utility, along with empirical validation on the space of Symmetric Positive Definite Matrices (SPDM).

**Strengths:**

* Novel Theoretical Contributions: First work to extend (ε,δ)-DP, GDP, and RDP to Hadamard manifolds in a unified framework. This significantly broadens the applicability of differential privacy to non-linear data.

* Computational Efficiency: The proposed exponential-wrapped mechanisms avoid computationally intensive MCMC methods, making them much more practical for real-world applications. The sampling procedures are straightforward and efficient.

* Improved Performance: The mechanisms show better utility than previous approaches, especially for small privacy budgets (ε). The Exponential-Wrapped Laplace mechanism requires only ∆/ε rate compared to 2∆/ε for nonhomogeneous manifolds in previous work.

**Weaknesses:**

Paper only uses the SPD manifold in experiments, but it would benefit from a more extensive empirical evaluation across different types of Hadamard manifolds and real-world applications. My main suggestion would be to extend the experiments to Hyperbolic spaces, which have a ton of interesting applications.

**Questions:**

NA

---

> ### Author Response · Authors · 2024-11-19
> **Reply to Weakness**
>
> Thank you for your thorough review and constructive feedback. We appreciate your insights, and we have addressed your concerns and questions in our response below:
>
> ## Weakness
>
>
> > Paper only uses the SPD manifold in experiments, but it would benefit from a more extensive empirical evaluation across different types of Hadamard manifolds and real-world applications. My main suggestion would be to extend the experiments to Hyperbolic spaces, which have a ton of interesting applications.
> \end{zitat}
>
> We agree that extending the experiments to hyperbolic spaces would be interesting as in recent years, hyperbolic space has been explored in many machine-learning areas such as hyperbolic reinforcement learning and hyperbolic neural networks. It's a direction we will for sure explore in future works.
>
> We have added a new experiment on a biomedical dataset, MedMNIST, as a real-world example of data in the SPDM space. See Appendix A.3.1 in the updated pdf for more details. We focus on outputting fr\'echet mean and achieving the GDP privacy guarantee. Similar to the simulation study, our EW Gaussian mechanism can achieve better utility for smaller privacy budgets.

---

### Official Review · Reviewer_tTEr · 2024-11-03

**Soundness:** 4
**Presentation:** 3
**Contribution:** 3
**Rating:** 5
**Confidence:** 3

**Summary:**

The paper proposes a method to extend Differentially Private mechanisms designed for Euclidean space to general Riemannian manifolds with non-negative sectional curvature, aka. Hadamard manifolds. The general idea is to use the Cartan-Hadamard theorem which says that for Hadamard manifolds, a universal cover (a map between two topological spaces) is diffeomorphic (i.e., bijective or invertible) to Euclidean space via the exponential map at any point. This exponential map on a Riemannian manifold is a way to "unroll" a point's local geometry, letting us move along geodesics (the "straightest" paths) from that point in the manifold, like traveling in a direction on a curved surface as if it were flat. Thanks to the invertibility of this exponential map, the paper proposes a straightforward way to perturb points in the Hadamard manifold to satisfy differential privacy, which they call *Exponential-wrapping* --- 1) first, map the point in the manifold to the Euclidean space using the inverse of the exponential map, 2) apply a well-known mechanism like Gaussian or Laplace, then 3) invert the noisy point back to the Hadamard manifold using the exponential map. The authors argue that this style of noise sampling is computationally efficient compared to other methods in literature that rely on MCMC methods. Additionally, the paper also provides a utility analysis for DP Frechet mean computed using their proposed exponentially-wrapped Gaussian and Laplace mechanisms. A simulation comparing prior MCMC methods with proposed mechanisms for DP Frechet mean is also presented.

**Strengths:**

- The proposed approach for randomizing outcomes in Hadamard manifolds is quite elegant and possibly easily extensible to other DP mechanisms.
- The paper provides utility analysis for the proposed mechanism.
- There's a clear computational advantage of the proposed mechanisms.
- The proposed exponentially-wrapped Gaussian mechanism could be useful in privatizing outcomes in high-dimensional manifolds.

**Weaknesses:**

- The main drawback of this approach is the cost that seems to be associated with the choice of the footpoint around which the mechanism is wrapped. More specifically, in equation (2) and (3), the utility for Frechet mean via the proposed mechanisms is closely tied to the choice of footpoint $p_0$ chosen on the manifold. Even when $p_0$ is set to be the center $m$ of the ball $B_r(m)$ in which all data-points lie, the utility upper bound is as high as $r$, which is of the same order as a random guess within the ball. This seems to be true for both the proposed EW Gaussian and EW Laplace mechanisms.
- Authors mention that the footpoint $p_0$ for their proposed mechanisms must be selected *a priori* without looking at the data. But, it isn't obvious to me why this footpoint can't be data-dependent. I suspect the utility of the mechanisms can significantly improve if some of the privacy budget is spent to privately determine a good footpoint for the mechanism first.
- Unlike Gaussian mechanism whose utility guarantee remains dimension independent, it's exponentially-wrapped counterpart proposed seem to be somehow dependent in the dimension size (eqn. (3)). I'm not certain if this dependence helps or hurts when dimension $d$ is large.
- The simulation results suggest that Riemannian Laplace mechanism proposed in the prior works performs better than this paper's EW Laplace in higher dimensions or in reasonable privacy regimes (1 > eps > 0.5). In lines 415-417, the authors acknowledge that this is expected as the utility advantage of their mechanism are reserved for non-homogeneous manifolds. Most manifolds we see in practice however are homogeneous---spheres, Torus, Grassmannian, etc.

**Questions:**

- If the selected footpoint is kept secret, only revealing the final perturbed result in the manifold, will there be a higher privacy risk if the choice of footpoint is influenced by the data? Could the authors give an example to help me see why the design of their mechanism fixes $p_0$ a-priori was necessary as opposed to setting $p_0$ to be the same as the unperturbed outcome?
- Could the authors describe the technical reason behind the dependence on dimension size for the EW-Gaussian mechanism in equation (3)? In lines 417-419, authors mention something about the volume distortions being more pronounced in higher dimensions, but I don't quite understand the magnitude of such distortions. Can they be avoided?

---

> ### Author Response · Authors · 2024-11-19
> **Reply to Weakness - Part 1**
>
> Thank you for your thorough review and constructive feedback. We appreciate your insights, and we have addressed your concerns and questions in our response below:
>
> ## Weakness
>
> > The main drawback of this approach is the cost that seems to be associated with the choice of the footpoint around which the mechanism is wrapped. More specifically, in equation (2) and (3), the utility for Frechet mean via the proposed mechanisms is closely tied to the choice of footpoint $p_0$ chosen on the manifold. Even when $p_0$ is set to be the center of the ball $B_r(m)$ in which all data-points lie, the utility upper bound is as high as $r$, which is of the same order as a random guess within the ball. This seems to be true for both the proposed EW Gaussian and EW Laplace mechanisms.
>
> We agree that the performance of the mechanism depends on the choice of the footpoint. However, there are ways to mitigate its impact. As you mentioned, one option is to allocate part of the privacy budget to identify a good footpoint. Alternatively, if prior information about the private data is available, it could be used to determine a suitable footpoint.
>
>
> > Authors mention that the footpoint $p_0$ for their proposed mechanisms must be selected a priori without looking at the data. But, it isn't obvious to me why this footpoint can't be data-dependent. I suspect the utility of the mechanisms can significantly improve if some of the privacy budget is spent to privately determine a good footpoint for the mechanism first.
>
>
> The footpoint $p$ determines the specific tangent space $T_p\mathcal{M}$ from which the push-forward distribution is applied. %Two Exponential-Wrapped Gaussian distributions with different footpoints indicate that they are the push-forward distribution of the (Euclidean) Gaussian distributions on two *different* tangent spaces. Demonstrating a mechanism satisfying some variant of differential privacy involves computing the ratio of two probability densities of the mechanism under two neighbouring datasets. If the footpoint is data-dependent, the ratio would be between two exponential-wrapped distributions with different footpoints. This makes the computation much more involved as it involves computing the ratio of the Jacobians and dealing with two norms on two different tangent spaces.
>
> Two Exponential-Wrapped Gaussian distributions with different footpoints represent the push-forward distributions of (Euclidean) Gaussian distributions on two distinct tangent spaces. To demonstrate that a mechanism satisfies a DP privacy guarantee, it is necessary to compute the ratio of the probability densities under two neighbouring datasets. If the footpoint is data-dependent, this ratio involves two Exponential-Wrapped distributions with different footpoints. This significantly complicates the computation, as it requires calculating the ratio of the Jacobians and managing two norms on two different tangent spaces. For more details, please refer to our response to the question section below.
>
> Spending part of the privacy budget to select a good footpoint can indeed improve utility in certain scenarios. Specifically, when there is no prior information about the data and the geodesic ball $B_r(m)$ containing the data has a large radius $r$, blindly choosing the center of the geodesic ball may not be ideal. In such cases, allocating a portion of the privacy budget to determine a suitable footpoint is the better approach. In fact, this is what we did in the newly added experiment on the real-world biomedical dataset, MedMNIST, as detailed in Appendix A.3.1.

---

> ### Author Response · Authors · 2024-11-19
> **Reply to Weakness - Part 2**
>
> > Unlike Gaussian mechanism whose utility guarantee remains dimension independent, it's exponentially-wrapped counterpart proposed seem to be somehow dependent in the dimension size (eqn. (3)). I'm not certain if this dependence helps or hurts when dimension $d$ is large.
>
> The utility guarantee of the Gaussian mechanism on $\mathbb{R}^d$ would still depend on its dimension $d$. This can be seen as
> \\[
> \int \|x - \eta\| f(x; \eta, \sigma^2) \, dx = \sqrt{2} \sigma^2 d  \frac{\Gamma((d+1)/2)}{\Gamma((d+2)/2)}
> \\]
> where $f(x; \eta, \sigma^2)$ denotes the multivariate gaussian density with mean $\eta$ and variance $\sigma^2 \mathbf{I}_d$ and $\Gamma$ denotes the gamma function. Similarly, for the Laplace mechanism, the dependence on $d$ can not be avoided either, see page 6 of [1].
>
>
> > The simulation results suggest that Riemannian Laplace mechanism proposed in the prior works performs better than this paper's EW Laplace in higher dimensions or in reasonable privacy regimes (1 $>$ eps $>$ 0.5). In lines 415-417, the authors acknowledge that this is expected as the utility advantage of their mechanism are reserved for non-homogeneous manifolds. Most manifolds we see in practice however are homogeneous---spheres, Torus, Grassmannian, etc.
>
>
> We agree that many manifolds encountered in practice are homogeneous. However, the homogeneity of a manifold is tied to its Riemannian metric. When a different Riemannian metric is applied to the same manifold, it may no longer remain homogeneous. For example, when the SPDM space is equipped with the Wasserstein/Procrustes metric, it is no longer homogeneous [2].

---

> > ### Comment · Reviewer_tTEr · 2024-11-23
> >
> > Thanks for the response! Yeah you're right, the utility cost of Gaussian mechanism on $R^d$ indeed linear in dimension. Based on the paper's expression (3), I'm curious if the dependence can be better/worse than linear in certain Hadamard manifolds. Could the authors comment on this?

---

> > > ### Author Response · Authors · 2024-11-23
> > >
> > > Sure. We can rewrite the expression (3) in a different form as the following,
> > > \begin{equation*}
> > > \sigma^2 d \frac{\Gamma((d+1)/2)}{\Gamma((d+2)/2)} \exp\left(-\frac{d(p_0, f(\mathcal{D}))^2}{2}\right) M\left(\frac{d+1}{2}, \frac{d}{2}, \frac{d(p_0, f(\mathcal{D}))^2}{2}\right) + d(p_0, f(\mathcal{D}))
> > > \end{equation*}
> > > where $M$ is the confluent hypergeometric function defined as follows,
> > > \\[
> > > M(a, b, x) = \sum_{n=0}^\infty \frac{a^{(n)}x^n}{b^{(n)}n!}
> > > \\]
> > > where $a^{(n)}$ is the rising fractorials defined as $a^{(n)} = a(a+1)\cdots (a+n-1)$ with $a^{(0)} = 1$.
> > > Note that $M(a, b, x) \ge e^x$ for $a \ge b$. Thus, we have
> > > \\[
> > > \sigma^2 d  \frac{\Gamma((d+1)/2)}{\Gamma((d+2)/2)} \exp\left(-\frac{d(p_0, f(\mathcal{D}))^2}{2}\right) M\left(\frac{d+1}{2}, \frac{d}{2}, \frac{d(p_0, f(\mathcal{D}))^2}{2}\right) \ge \sigma^2 d \frac{\Gamma((d+1)/2)}{\Gamma((d+2)/2)}
> > > \\]
> > > Therefore, the dependence of $d$ is worse on Hadamard manifolds. Moreover, the dependence of $d$ is affected by the value of $d(p_0, f(\mathcal{D}))$. If we have $d(p_0, f(\mathcal{D})) = 0$, then expression (3) reduces to the Euclidean cases,
> > > \\[
> > > \sigma^2 d \frac{\Gamma((d+1)/2)}{\Gamma((d+2)/2)}.
> > > \\]
> > > We hope this answers your question.

---

> > > > ### Author Response · Authors · 2024-11-27
> > > > **Follow-Up on Your Feedback**
> > > >
> > > > Thank you for your previous feedback. We hope our response addressed your concerns effectively. Please let us know if there is anything else we can clarify or improve upon. We would greatly appreciate any further thoughts you might have regarding our submission.

---

> ### Author Response · Authors · 2024-11-19
> **Reply to Questions**
>
> ## Questions
>
>
> > If the selected footpoint is kept secret, only revealing the final perturbed result in the manifold, will there be a higher privacy risk if the choice of footpoint is influenced by the data? Could the authors give an example to help me see why the design of their mechanism fixes $p_0$ a-priori was necessary as opposed to setting $p_0$ to be the same as the unperturbed outcome?
>
>
> Yes, there will be a higher privacy risk if the footpoint is selected based on the data. To understand why the footpoint needs to be selected a-priori, let's consider the EW Laplace mechanism where the footpoint is chosen to be the unperturbed outcome $f(\mathcal{D})$. The density of the mechanism is then given below,
>
> \\[
> g(y) \propto \frac{1}{J_{f(\mathcal{D})}(\log_{f(\mathcal{D})}(y))} \exp \left(-\frac{||\log_{f(\mathcal{D})}(y)||_{f(\mathcal{D})}}{\sigma} \right),
> \\]
>
> To show it satisfies pure DP, it amounts to show the ratio of the two densities (one density corresponding to 1 neighbouring dataset) is smaller than $e^\varepsilon$. The ratio will have the following form,
>
> \\[
> \frac{J_{f(\mathcal{D}^{\prime})}(\log_{f(\mathcal{D}^{\prime})}(y))}{J_{f(\mathcal{D})}(\log_{f(\mathcal{D})}(y))} \exp \left(\frac{||\log_{f(\mathcal{D}^{\prime})}(y)||\_{f(\mathcal{D}^{\prime})} - ||\log\_{f(\mathcal{D})}(y)||_{f(\mathcal{D})}}{\sigma} \right),
> \\]
>
> This ratio is difficult to evaluate because: 1) the Jacobian terms can no longer be cancelled out, making them challenging to compute, and 2) the two norms within the $\exp$ are defined on different (tangent) spaces, meaning the usual triangle inequality cannot be applied. However, if we select the footpoint a priori, these two issues can be avoided.
>
>
> > Could the authors describe the technical reason behind the dependence on dimension size for the EW-Gaussian mechanism in equation (3)? In lines 417-419, authors mention something about the volume distortions being more pronounced in higher dimensions, but I don't quite understand the magnitude of such distortions. Can they be avoided?
>
> As mentioned above, the dependence of the dimension $d$ can be seen from the following equation
>
> \\[
> \int \|x - \eta\| f(x; \eta, \sigma^2) \, dx = \sqrt{2} \sigma^2 d  \frac{\Gamma((d+1)/2)}{\Gamma((d+2)/2)}.
> \\]
>
> The volume distortions can be reduced if a good footpoint is selected. It can be avoided if the footpoint is selected to be exactly the unperturbed outcome, but this is not possible as I explained above.
>
>
> ## References
>
> [1]: Matthew Reimherr, Karthik Bharath, and Carlos Soto. Differential privacy over Riemannian manifolds. In Advances in Neural Information Processing Systems, volume 34, pp. 12292–12303. Curran Associates, Inc., 2021.
>
> [2]: Yann Thanwerdas and Xavier Pennec. $O(n)$-invariant Riemannian metrics on spd matrices. Linear Algebra and its Applications, 661:163–201, 2023. ISSN 0024-3795. doi: https://doi.org/10.1016/j.laa. 2022.12.009.

---

### Official Review · Reviewer_Yq1S · 2024-11-05

**Soundness:** 2
**Presentation:** 2
**Contribution:** 2
**Rating:** 6
**Confidence:** 2

**Summary:**

This paper proposed DP and approximate DP mechanisms when the function f that we want to privatize falls into a Hadamard Manifold. In particular, the paper proposed to use Exponential-Wrapped Laplace Mechanism to achieve eps-DP and Exponential-Wrapped Gaussian Mechanism to achieve (eps,delta)-DP.

**Strengths:**

The paper proposed simple and clean DP mechanisms when the value function falls into the Hadamard Manifold. Simulation results also show good privacy / accuracy tradeoffs.

**Weaknesses:**

From a technical point of view, the biggest concern is that the paper made a strong assumption that the values of the function fall into a manifold with good properties. With these properties, one can directly apply sampling techniques via Exponential-Wrapped Probability to achieve DP or approximate DP. The techniques become a bit straight-forward.

The simulation is only for synthetic data, it is not clear whether there are any real use cases / problems that fall in the case that value functions are in such a good manifold.

**Questions:**

Are there any practical problems in the real world such that the values have fallen into Hadamard Manifolds?

---

> ### Author Response · Authors · 2024-11-19
> **Reply to Weakness**
>
> Thank you for your thorough review and constructive feedback. We appreciate your insights, and we have addressed your concerns and questions in our response below:
>
> ## Weakness
>
> > From a technical point of view, the biggest concern is that the paper made a strong assumption that the values of the function fall into a manifold with good properties. With these properties, one can directly apply sampling techniques via Exponential-Wrapped Probability to achieve DP or approximate DP. The techniques become a bit straight-forward.
>
> SPD matrices play a significant role in medical imaging and are often called tensors when modelling the covariance matrix of water's Brownian motion in Diffusion Tensor Imaging [1, 2]. They have also been used in shape analysis to capture joint variability across different locations [3, 4]. Furthermore, SPD matrices are widely applied in computer vision and image analysis tasks, including segmentation, grouping, and motion analysis [5, 6, 7, 8]. For a summary of the practical applications of SPDM space, refer to Chapter 3 of [9].
>
> Hyperbolic spaces have gained considerable attention in machine learning for their ability to enhance performance and efficiency, particularly in learning representations of hierarchical structures and graph data [10, 11]. A key characteristic of hyperbolic space is its exponential volume growth, which allows tree-like hierarchical data to be embedded with low distortion while requiring only a few dimensions [12]. It leads to lower complexity, fewer parameters, and less overfitting. For more details, refer to [13].
>
> We will include more background materials like the above in the paper for the reader's information.
>
> > The simulation is only for synthetic data, it is not clear whether there are any real use cases / problems that fall in the case that value functions are in such a good manifold.
>
> We have added a new experiment on a real-world biomedical dataset, MedMNIST. Please refer to Appendix A.3.1 in the updated pdf for more details. This experiment focuses on computing the Fréchet mean while ensuring a GDP privacy guarantee. Similar to the simulation study, our Exponential-Wrapped Gaussian mechanism demonstrates better utility for smaller privacy budgets.
>
> ## References
>
> [1]: Peter J. Basser, James Mattiello, Denis Le Bihan, MR diffusion tensor spectroscopy and imaging, Biophysical Journal 66 (1994) 259–267.
>
> [2]: Denis Le Bihan, Jean-François Mangin, Cyril Poupon, C.A. Clark, S. Pappata, N. Molko, H. Chabriat, Diffusion tensor imaging: concepts and applications, Journal of Magnetic Resonance Imaging 13 (4) (2001) 534–546.
>
> [3]: Pierre Fillard, Vincent Arsigny, Xavier Pennec, Paul M. Thompson, Nicholas Ayache, Extrapolation of sparse tensor fields: application to the modelling of brain variability, in: Gary Christensen, Milan Sonka (Eds.), Proc. of Information Processing in Medical Imaging 2005 (IPMI’05), Glenwood Springs, Colorado, USA, in: LNCS, vol. 3565, Springer, July 2005, pp. 27–38, PMID: 17354682.
>
> [4]: Pierre Fillard, Vincent Arsigny, Xavier Pennec, Kiralee M. Hayashi, Paul M. Thompson, Nicholas Ayache, Measuring brain variability by extrapolating sparse tensor fields measured on sulcal lines, NeuroImage 34 (2) (January 2007) 639–650, also as INRIA Research Report 5887, April 2006, PMID: 17113311.
>
> [5]: Gérard Medioni, Mi-Suen Lee, Chi-Keung Tang, A Computational Framework for Segmentation and Grouping, Elsevier, 2000.
>
> [6]: Thomas Brox, Johachim Weickert, Bernhard Burgeth, Pavel Mrázek, Nonlinear structure tensors, Image and Vision Computing 24 (1) (2006) 41–55.
>
> [7]: Johachim Weickert, Thomas Brox, Diffusion and regularization of vector- and matrix-valued images, in: M.Z. Nashed, O. Scherzer (Eds.), Inverse Problems, Image Analysis, and Medical Imaging, in: Contemporary Mathematics, vol. 313, AMS, Providence, 2002, pp. 251–268.
>
> [8]: Joachim Weickert, Hans Hagen (Eds.), Visualization and Processing of Tensor Fields. Mathematics and Visualization, Springer, 2006.
>
> [9]: Xavier Pennec, Stefan Sommer, and Tom Fletcher. Riemannian geometric statistics in medical image analysis. Academic Press, 2019.
>
> [10]: Maximillian Nickel and Douwe Kiela. Poincare embeddings for learning hierarchical representations. Advances in neural information processing systems, 30, 2017.
>
> [11]: Benjamin P Chamberlain, James Clough, and Marc P Deisenroth. Neural embeddings of graphs in hyperbolic space. arXiv preprint arXiv:1705.10359, 2017.
>
> [12]: Rik Sarkar. Low distortion delaunay embedding of trees in hyperbolic plane. In International Symposium on Graph Drawing, pp. 355–366. Springer, 2011.
>
> [13]: Edoardo Cetin, Benjamin Paul Chamberlain, Michael M. Bronstein, and Jonathan J Hunt. Hyperbolic deep reinforcement learning. In The Eleventh International Conference on Learning Representations, 2023.

---

> ### Author Response · Authors · 2024-11-19
> **Rely to Questions**
>
> ## Questions
>
> > Are there any practical problems in the real world such that the values have fallen into Hadamard Manifolds?
>
> As mentioned above, the two prominent Hadamard manifolds—hyperbolic space and SPDM spaces—have plenty of real-world applications.

---

> ### Author Response · Authors · 2024-11-30
> **Follow-Up on Your Feedback**
>
> Thank you for your detailed review and constructive feedback. We are confident that our response has addressed your concerns. If there are any additional points requiring clarification or refinement, please let us know. We would greatly value any additional thoughts you may have on our submission.

---

> > ### Comment · Reviewer_Yq1S · 2024-12-01
> >
> > Thank you for the response, it did address my concerns. I have raised my score accordingly.

---

> > > ### Author Response · Authors · 2024-12-01
> > >
> > > Thank you for your feedback and for increasing the score. We appreciate your time and consideration.

---

### Official Review · Reviewer_w4DG · 2024-11-07

**Soundness:** 3
**Presentation:** 2
**Contribution:** 3
**Rating:** 6
**Confidence:** 3

**Summary:**

This paper extends the Differential Privacy framework to the class of complete and simply connected Riemannian manifolds - Hadamard manifolds. The major contributions are the first extension of Laplace and Gaussian mechanisms that could achieve DP, GDP and RDP on these manifolds. The new method is named Exponential-Wrapped mechanism in the sense that they are deducted through the Exponential Mapping  and their push forward functions. The approach gives efficient, straightforward algorithms comparable to the vanilla mechanisms in Euclidean space. We further demonstrate the effectiveness of our methodology through simulations on the space of Symmetric Positive Definite Matrices, a frequently used Hadamard manifold in statistics. In numerical experiments, the Exponential-Wrapped mechanisms beat traditional MCMC-based approaches in efficiency and achieve comparable utility to the Riemannian Laplace mechanism with enhanced utility.

**Strengths:**

The paper gives the first extension of the common Laplace and Gaussian mechanisms in the context of more generalized Riemannian Geometry settings than the common multi-dimensional Euclidean space setting, so in that sense the contribution's importance and its novelty are both straight-forward.

Although the formulation becomes more intricate, the steps that the authors took to develop these mechanisms seem well-established to me.

I'm not very familiar with the exact applications of such geometric settings, but judging from my limited knowledge the findings could be useful.

**Weaknesses:**

To me the major issue is the writing of the paper. It could be that my geometry is simply rusty, but I'm not sure how familiar the general community for this venue are with the math concepts used in this paper. In light of that, I think this paper sort of takes it for granted that people know all these things (such as Exponential mapping, push-forward, etc.) already. I find the preliminary section to be unsatisfying and insufficient of the crucial math concepts used later in the deduction of the mechanisms in general. My (biased) opinion is that some of these things deserve more explanation before we go deeper into the weeds.

I was also slightly surprised that when $\varepsilon$ is about [0.2, 0.6], the benchmark algorithm performs better than the proposed algorithm. I think in Euclidean spaces at least for pure DP the Laplace mechanism should be pretty close to being optimal for such small $\varepsilon$ values?

There isn't much theoretical characterization of how good such generalization mechanisms are, even in the pure DP case. But perhaps that's because the community simply don't know much in this more complicated context.

**Questions:**

Do you have any insights into if there could be similar extensions for other traditional mechanisms such as the K-norm mechanism (Laplace is a special case of K-norm using the $\ell_1$-norm)?

I'm not very familiar with Riemannian Geometry and Manifolds, to me it is surprising that the extended version of Laplace has additional parameter $p_0$ (the footprint) compared to the Euclidean space version. I thought the Hardamard Manifold is basically a warped version of Euclidean space, but then in principle why do we need to use an additional $p_0$ to define the Laplace Mechanism? To help the understanding, can the authors recast the results back into Euclidean space and show the value of all these parameters given $\epsilon$ for pure DP budget?

Minor:

In line 108 the sentence seems incomplete: "There exists a unique [???] starting from..."

The notion $p_0$ abruptly appeared in line 188.

---

> ### Author Response · Authors · 2024-11-19
> **Reply to Weakness**
>
> Thank you for your thorough review and constructive feedback. We appreciate your insights, and we have addressed your concerns and questions in our response below:
>
> ## Weakness
>
> > To me the major issue is the writing of the paper. It could be that my geometry is simply rusty, but I'm not sure how familiar the general community for this venue are with the math concepts used in this paper. In light of that, I think this paper sort of takes it for granted that people know all these things (such as Exponential mapping, push-forward, etc.) already. I find the preliminary section to be unsatisfying and insufficient of the crucial math concepts used later in the deduction of the mechanisms in general. My (biased) opinion is that some of these things deserve more explanation before we go deeper into the weeds.
>
> We agree that the differential geometry stuff can be hard to get into without a proper introduction. We will reorganize the background material section for Riemannian geometry and provide more intuitive explanations to help the readers grasp these concepts better.
>
> > I was also slightly surprised that when $\varepsilon$ is about [0.2, 0.6], the benchmark algorithm performs better than the proposed algorithm. I think in Euclidean spaces at least for pure DP the Laplace mechanism should be pretty close to being optimal for such small $\varepsilon$ values?
>
> For pure DP, both our proposed algorithm and the benchmark algorithm can be viewed as variants of the Laplace mechanism. The benchmark algorithm adapts the Laplace distribution from Euclidean spaces to manifolds by replacing the $l_1$ -norm with the Riemannian distance function. In contrast, our Exponential-Wrapped Laplace mechanism extends the Laplace distribution to manifolds by leveraging the push-forward distribution of the (Euclidean) Laplace distribution on a tangent space $T_p\mathcal{M} \simeq \mathbb{R}^d$.
>
> The key advantage of our algorithm over the benchmark lies in its significantly faster computational performance. Additionally, for non-homogeneous manifolds, our mechanism uses a reduced rate of $\Delta/\varepsilon$ compared to the benchmark's $\Delta/\varepsilon$. On homogeneous manifolds, however, both algorithms operate with a rate of $\Delta/\varepsilon$. Given that the SPDM space in the simulation study is homogeneous, we do not expect our algorithm to provide better utility than the benchmark in this scenario.
>
> > There isn't much theoretical characterization of how good such generalization mechanisms are, even in the pure DP case. But perhaps that's because the community simply don't know much in this more complicated context.
> \end{zitat}
>
> In Theorem 4.1, we provide upper bounds on the expected distance between the output of our Exponential-Wrapped mechanisms and the unperturbed outcome. While these bounds are not tight due to the reliance on the triangle inequality, they offer valuable insights into how the mechanism's performance depends on various parameters, including $n, d, \mu, r$.

---

> ### Author Response · Authors · 2024-11-19
> **Reply to Questions**
>
> ## Questions
>
> > Do you have any insights into if there could be similar extensions for other traditional mechanisms such as the K-norm mechanism (Laplace is a special case of K-norm using the $l_1$-norm)?
>
> The $K$-norm mechanism has been extended to general manifolds in [1].
>
> > I'm not very familiar with Riemannian Geometry and Manifolds, to me it is surprising that the extended version of Laplace has additional parameter $p_0$ (the footprint) compared to the Euclidean space version. I thought the Hardamard Manifold is basically a warped version of Euclidean space, but then in principle why do we need to use an additional $p_0$ to define the Laplace Mechanism? To help the understanding, can the authors recast the results back into Euclidean space and show the value of all these parameters given $\varepsilon$ for pure DP budget?
>
> Intuitively, a manifold is a space that is locally Euclidean. In other words, if we focus on a small region of the manifold, it behaves like an Euclidean space. However, when we zoom out and observe the manifold as a whole, its characteristics can differ significantly from those of Euclidean space. For example, on a sphere, parallel lines converge and eventually meet, while in hyperbolic space (a Hadamard manifold), they diverge and spread apart.
>
> As mentioned above, our Exponential-Wrapped Laplace mechanism extends the Laplace mechanism to manifolds by leveraging the push-forward distribution of the (Euclidean) Laplace distribution on a tangent space $T_p\mathcal{M} \simeq \mathbb{R}^d$. However, each point $p$ on the manifold has a corresponding tangent space $T_p\mathcal{M}$ and the footpoint $p$ indicates the specific tangent space $T_p\mathcal{M}$ from which the push-forward distribution is applied.
>
> Our Exponential-Wrapped Laplace mechanism (similarly for the Gaussian mechanism) can be described through the following steps:
>
> 1. Select a footpoint $p_0$.
> 2. Use the $\log_{p_0}$ function to project the unperturbed outcome $f(\mathcal{D}) \in \mathcal{M}$ to the tangent space $T_{p_0}\mathcal{M}$ to obtain $\log_{p_0}(f(\mathcal{D})) \in T_{p_0}\mathcal{M} \simeq \mathbb{R}^d$.
> 3. Perturb $\log_{p_0}(f(\mathcal{D}))$ use the usual Euclidean Laplace mechanism to obtain $v = \log_{p_0}(f(\mathcal{D})) + \text{Laplace}(0, \sigma)$ where $\sigma = \Delta / \varepsilon$.
> 4. Use the $\exp_{p_0}$ to project $v$ back to the manifold to obtain the final perturbed outcome $\exp_{p_0}(v)$.
> \end{enumerate}
>
> Hopefully, this helps you better understand our mechanisms.
>
> **References**:
>
> [1]: C. Soto, K. Bharath, M. Reimherr, and A. Slavkovic. Shape and structure preserving differential privacy. In Advances in Neural Information Processing Systems, volume 35, pages 24693–24705. Curran Associates, Inc., 2022.
>
> [2]: Matthew Reimherr, Karthik Bharath, and Carlos Soto. Differential privacy over Riemannian manifolds. In Advances in Neural Information Processing Systems, volume 34, pp. 12292–12303. Curran Associates, Inc., 2021.

---

> ### Author Response · Authors · 2024-11-19
> **Reply to Minor**
>
> ## Minor
>
> Thanks for pointing out the typos. The typos have been fixed in the updated pdf.
>
> > In line 108 the sentence seems incomplete: In line 108 the sentence seems incomplete: "There exists a unique [???] starting from..."
>
> It should be "There exists a unique geodesic $\gamma_{(p, v)}(t)$ starting from..."
>
> > The notion $p_0$ abruptly appeared in line 188.
>
> It should be $p$ instead of $p_0$.

---

> ### Author Response · Authors · 2024-11-30
> **Follow-Up on Your Feedback**
>
> Thank you for your detailed review and constructive feedback. We are confident that our response has addressed your concerns. If there are any additional points requiring clarification or refinement, please let us know. We would greatly value any additional thoughts you may have on our submission.

---

> > ### Comment · Reviewer_w4DG · 2024-12-02
> > **Thank you for the response**
> >
> > I appreciate your extensive reply. They have indeed addressed my concerns! I will keep my current score.

---

### Meta-Review · Area_Chair_qZEV · 2024-12-20

**Metareview:**

This paper provides results on differentially private release of data living on Hadamard manifolds, which includes, e.g., the private release of SPSD covaraince matrices. The reviewers felt that this paper had technical merit, and all major technical concerns were address by the authors during the response period. Ultimately, however, the topic of the paper is narrow, so perhaps better suited to a more specialized venue focused on differential privacy. The paper could have been strengthened with applications and real-data experiments that justify studying this problem in full generality. To their credit, the authors provided one such experiment during the response period, but their new method does not convincingly outperform the prior work. We hope that based on the reviewer feedback, the authors are able to improve their work and refine its presentation for submission to another venue.

**Additional Comments On Reviewer Discussion:**

See main metareview where this is discussed.

---

### Decision · Program_Chairs · 2025-01-22

Reject